# Plug-and-Play Label Map Diffusion for Universal Goal-Oriented Navigation

## Abstract

In embodied vision, Goal-Oriented Navigation (GON) requires robots to locate a specific goal within an unexplored environment. The primary challenge of GON arises from the need to construct a Bird's-Eye-View (BEV) map to understand the environment while simultaneously localizing an unobserved goal. Existing map-based methods typically employ self-centered semantic maps, often facing challenges such as reliance on complete maps or inconsistent semantic association. To this end, we propose Plug-and-Play Label Map Diffusion (PLMD), which defines a novel map completion diffusion model based on Denoising Diffusion Probabilistic Models (DDPM). PLMD generates obstacle and semantic labels for unobserved regions through a diffusion-based completion process, thereby enabling goal localization even in partially observed environments. Moreover, it mitigates inconsistent semantic association by leveraging structural consistency between known and unknown obstacle layouts and integrating obstacle priors into the semantic denoising process. By substituting predicted labels for unobserved regions, robots can accurately localize the specified objects. Extensive experiments demonstrate that PLMD **(I)** effectively expands the region of unknown maps, **(II)** integrates seamlessly into existing navigation strategies that rely on semantic maps, **(III)** achieves state-of-the-art performance on three GON tasks.

## 1 Introduction

In Goal-Oriented Navigation (GON) tasks, robots are placed in an unknown indoor environment and tasked with navigating to a user-specified category of objects (e.g., a bed) or instance image based on visual observations. Depending on the type of goal and the number of robots, GON can be divided into several subgenres, and we focus on ObjectNav (ON) (Chaplot et al., 2020b; Zhou et al., 2023; Yu et al., 2023b), Instance-ImageNav (IIN) (Krantz et al., 2022; 2023; Lei et al., 2024), and Multi-Robot ObjectNav (MRON) (Yu et al., 2023a; Shen et al., 2024). Since the robots start in an unfamiliar environment, previous methods (Chaplot et al., 2020b; Du et al., 2020; Krantz et al., 2023; Ramakrishnan et al., 2022; Wang et al., 2022; Lei et al., 2024) typically construct Semantic Bird-Eye-View (BEV) map for cognitive memory of the environment. To address the incomplete visibility of the map, prior works rely on either end-to-end reinforcement learning (RL) (Zhu et al., 2017; Wortsman et al., 2019; Maksymets et al., 2021; Du et al., 2021; Mayo et al., 2021; Ye & Yang, 2021) or modular approaches (Chaplot et al., 2020b;a;c; Krantz et al., 2022; 2023; Lei et al., 2024). RL-based methods attempt to directly learn goal-directed exploration policies, while modular methods build semantic maps to infer plausible goal location by leveraging semantic relationships (e.g., tables and chairs often co-occur). However, these approaches typically rely on complete maps, making them effective when the map is fully observed but unreliable when reasoning about unobserved map regions.

To overcome this challenge, a natural solution is to employ generative models that can complete missing parts of semantic maps. Recently, Denoising Diffusion Probabilistic Models (DDPMs) (Ho et al., 2020; Song et al., 2020) have achieved remarkable progress in semantic generation and image inpainting (Lugmayr et al., 2022; Luo et al., 2023; Wang et al., 2022; Zhu et al., 2023; Liu et al., 2024), effectively addressing the challenge of completing unknown regions in semantic maps. Specifically, (Ji et al., 2024; Li et al., 2025) attempts to constrain the diffusion model to learn the statistical distribution patterns of objects in semantic maps, enabling it to generate the goal's location more effectively. However, it focuses heavily on the correlations between scene object semantics while

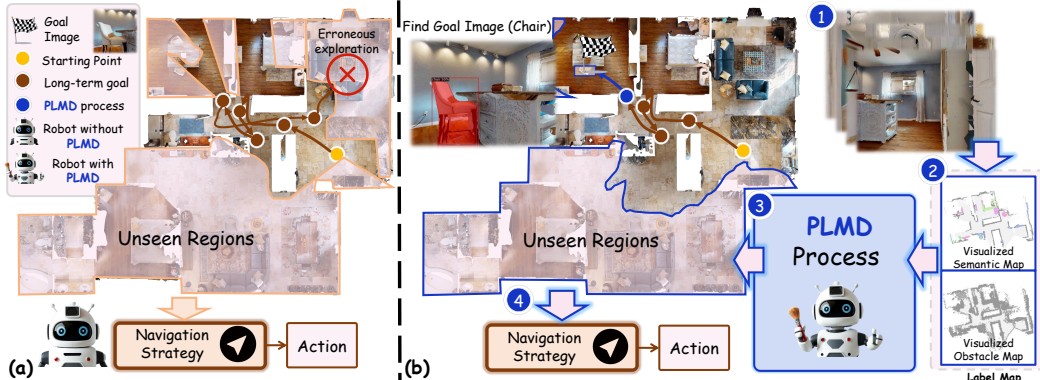

Figure 1: Different implementations of map-dependent GON tasks. **(a)** The original map-based navigation strategy. **(b)** Our PLMD is able to extend the semantic and obstacle information of the unseen map regions without re-training.

neglecting the consistent semantic association between known and unknown map obstacles. Unlike natural images, BEV maps contain large areas of free space and sparse object pixels. This leads to issues such as room boundary drift and semantic hallucinations in unobserved regions when learning only semantic associations. In addition, the structure of maps has been proven to be more beneficial for semantic consistency in the early stages of diffusion denoising (Liu et al., 2024). Therefore, we leverage known obstacles and object semantic information to rebuild unknown regions, while constructing a **Label Map** that explicitly encodes obstacles and scene object categories. This process enables navigation strategies to capture contextual environmental relationships (e.g., interactions between objects and potential obstacles), thereby providing multi-level semantic and structural details at the label granularity.

In this paper, we propose Plug-and-Play Label Map Diffusion (PLMD), designed to seamlessly integrate with any GON strategy. As illustrated in Fig. 1, compared to the navigation strategy using the original map, our PLMD-assisted navigation strategy is divided into four steps. **(I)** Robot constructs egocentric semantic and obstacle map representations to record both object semantics and obstacle distributions in the observed scene. **(II)** Representations are concatenated into composite label maps, encoded with distinct pixel values for visualization, after which the semantic and obstacle regions corresponding to unobserved areas are masked. **(III)** We utilize obstacle maps to drive the semantic map denoising process and monitor the semantic association through resampling iterations. During PLMD training, obstacle-aware feature modulation provides stable structural constraints during early semantic diffusion steps when sparse semantic signals are weakest, thereby preventing physically implausible room geometries. **(IV)** Following PLMD generation of the prediction map, we employ a clustering algorithm (Campello et al., 2013) to identify potential navigation goals (i.e., pixel clusters matching the goal color) in the new map, forming a candidate goal set. PLMD continuously generates predicted maps based on the updated label map until the candidate goal set is identified or the navigation strategy locates the goal.

We present the following contributions: ❶ *Label-level Map Completion.* We propose PLMD, which operates at the granularity of labels rather than global map structures, enabling fine-grained completion of unobserved regions. ❷ *Obstacle-guided Denoising.* We design a label-guided denoising process that leverages obstacle distributions as structural constraints, ensuring consistent and reliable semantic reconstruction at the pixel level. ❸ *Extensive Validation.* We evaluate our PLMD using multiple navigation strategy baselines in the realistic 3D environments of Habitat-Matterport3D (HM3D) (Ramakrishnan et al., 2021; Yadav et al., 2023b) and Matterport3D (MP3D) (Chang et al., 2017). Experimental results demonstrate the effectiveness of the algorithm in assisting navigation strategies to locate the goal.

## 2 RELATED WORK

**Goal-Oriented Navigation.** Goal-Oriented Navigation (GON) tasks can be broadly categorized into two main approaches: end-to-end and modular methods. The former primarily employs reinforcement

learning (RL) (Wortsman et al., 2019; Zhang et al., 2022; Zhu et al., 2017) or imitation learning (IL) (Ramrakhya et al., 2022; 2023) to learn navigation policies. Specifically, they attempt to encode visual observations into latent codes and predict low-level actions (Wijmans et al., 2019), learn visual representations (Khandelwal et al., 2022; Kotar et al., 2023; Mayo et al., 2021), learn historical state representations (Du et al., 2023), adopt auxiliary tasks (Ye et al., 2021), or use data augmentation to improve navigation performance (Deitke et al., 2022; Maksymets et al., 2021). However, this implicit encoding of unknown 3D scenes and direct prediction of actions suffer from low sample efficiency and difficulty in capturing fine-grained semantic context. To address these issues, modular methods (Krantz et al., 2022; Yu et al., 2023b; Zhou et al., 2023; Yu et al., 2023a; Krantz et al., 2023; Kuang et al., 2024; Zhang et al., 2024; Shen et al., 2024; Lei et al., 2024; Yin et al., 2025a) typically map visual perceptions to top-down semantic maps or 3D maps and update them online. Some approaches utilize Large Language Models (LLMs) (Achiam et al., 2023; Touvron et al., 2023) to select long-term goal points on semantic maps, or employ supervised or self-supervised learning to learn goal-related semantic associations. Based on online-constructed semantic maps, (Yu et al., 2023b) and (Zhou et al., 2023) leverage large language models for navigation decision-making, (Ramakrishnan et al., 2022) predicts the nearest frontier to the goal, and (Zhai & Wang, 2023) predicts the absolute coordinates of the goal. However, they all rely on the completeness of semantic maps. Alternatively, we propose a plug-and-play diffusion model that leverages obstacle-guided denoising to complete unobserved regions in maps, thereby enhancing goal localization for universal navigation.

**Map Prediction for GON.** Recent studies have attempted to predict unknown regions of semantic maps. For example, (Georgakis et al., 2021) adds semantic predictions using a two-stage segmentation model. (Liang et al., 2021) implicitly encodes predicted semantic maps into an RL-based policy. (Zhang et al., 2024) learns contextual semantic relationships in the map to determine the goal location. In parallel, several works have explored leveraging pre-trained models for semantic mapping and navigation. (Chen et al., 2022) proposes a weakly-supervised multi-granularity map to represent both semantics and fine-grained attributes, while (Wang et al., 2023) constructs a dynamically growing grid memory map with instruction-relevant aggregation. More recently, (Guo et al., 2025) introduces incremental 3D Gaussian localization for image-goal navigation, enabling accurate goal pose estimation via differentiable rendering. (Li et al., 2025) achieves semantic prediction by distilling spatial prior knowledge from LLMs into generative stream models. Notably, unlike previous work, we do not leverage PLMD as a navigation strategy but as a map prediction tool to assist navigation, capable of collaborating with any map-based navigation strategy.

**Diffusion Models for Image Inpainting.** Image Completion (Lugmayr et al., 2022; Luo et al., 2023; Wang et al., 2022; Zhu et al., 2023; Liu et al., 2024) based on DDPMs(Ho et al., 2020; Song et al., 2020) has been proven to be an effective learning approach. (Lugmayr et al., 2022) modifies reverse diffusion iterations by sampling from the unmasked regions of an image, (Luo et al., 2023) proposes a stochastic differential equation (SDE) method for general image restoration without relying on prior knowledge, (Wang et al., 2022) extends image inpainting to different degradation operators using a zero-shot framework, and (Liu et al., 2024) mitigates semantic discrepancies through structure guidance. Currently, research on map completion for embodied navigation is limited, with only (Ji et al., 2024) utilizing the diffusion model to generate goal pixels in semantic maps. Our PLMD drives the semantic denoising process using obstacle maps, leveraging the semantic consistency between known and unknown regions of the map to estimate denoising targets and generate label map vectors for unknown regions.

## 3 METHOD

### 3.1 PRELIMINARIES: LABEL MAP CONSTRUCTION FOR EMBODIED NAVIGATION

To facilitate goal-oriented search and localization, most embodied navigation approaches rely on egocentric semantic maps. Specifically, at each time step, the robots' RGB-D observations and pose information are acquired. Subsequently, depth images are used to project each pixel along with its semantic label into 3D space. Points within a predefined height range relative to the robot are designated as occupied. These points are then discretized into a voxel occupancy grid and integrated along the height dimension to construct an egocentric map. This egocentric representation is transformed to a geocentric coordinate frame based on the robot's pose and fused with previously

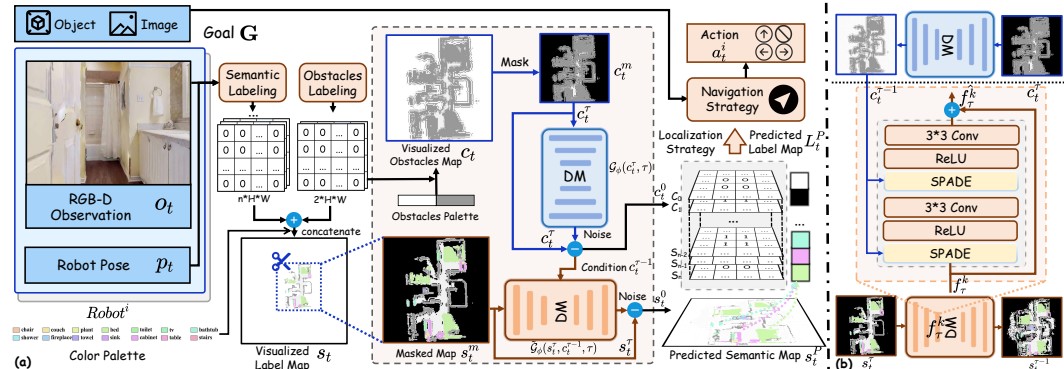

Figure 2: Framework of our PLMD. **(a)** illustrates the structure of the PLMD, showing the navigation process guided by the predicted label map. 'DM' stands for Diffusion Model. **(b)** shows the obstacle-aware feature modulation network for the semantic map network.

existing global semantic labels. The resulting output is a map $M_t \in \mathbb{R}^{(n+4) \times H \times W}$, where $H$ and $W$ denote spatial dimensions, while $n+4$ represents the total channel count. The channels consist of: (1) $n$ semantic class maps, (2) an occupancy map (occupied regions), (3) a free-space map (unoccupied regions), and (4) current/past position indicators of the robot.

We represent the environment with **Label Maps**, formed by merging **Obstacle Maps** (constructed from occupancy and free-space information) and **Semantic Maps**, and rendered using fixed color palettes. Since label maps only incorporate partial observations from current and historical viewpoints, robots face challenges in anticipating semantic and obstacle distribution structures of unexplored areas. When encountering occluded environments, robots can infer unobserved surrounding environments based on learned associations between scene objects and obstacles.

### 3.2 LABEL MAP DIFFUSION POLICY

In this section, we describe the training methodology for the diffusion model used in generating visualized label map.

**Label Map Data Collection.** To ensure the rationality of the data, we generate training and validation data from label maps collected during the interaction of robot with the environment. First, we randomly initialize a starting position within indoor environments (*For convenience, all data examples here use HM3D_v0.1. Details regarding MP3D are provided in Appendix A.6.*). Since our diffusion model operates on visualized maps, we employ the mapping process described in Section 3.1 to construct the observation pair $\mathcal{O} = \{(s_t, s_t^m, c_t, c_t^m)\}$, where $s_t$ denotes the semantic map at time step $t$, $s_t^m$ represents the masked semantic map (with unexplored regions indicated by mask $m$), $c_t$ correspond to the obstacle map and $c_t^m$ represents masked map of $c_t$. To learn generalizable priors for scene completion, we pre-train the PLMD on a dedicated dataset collected separately from the navigation evaluation environments. Specifically, the robot employs the Frontier Exploration Strategy (FBE) (Yamauchi, 1997) to navigate from multiple initial positions, executing up to $F = 500$ navigation steps. An incomplete label maps is stored every 25 steps. The fully explored semantic map $s_{gt}$ and obstacle map $c_{gt}$ at the $F$th step form the training label map observation pair $\mathcal{O}_{label} = \{(s_{gt}, s_t^m, c_{gt}, c_t^m)\}$. We define the collection of label map observation pairs that have executed all predefined episodes ($\mathcal{N}$ (2,000)) as the label map dataset: $\mathcal{A} = \sum_{a=1}^{\mathcal{N}} (\bigcup_{t \in \{0,25,50,...,F\}} \{(s_{gt}, s_t^m, c_{gt}, c_t^m)\})_a$. *Details of the training set $\mathcal{T}$ and validation set $\mathcal{V}$ of $\mathcal{A}$ are given in Appendix A.5.*

**Diffusion Training Process.** Given the full semantic map $s_{gt} \in \mathbb{R}^{3 \times H \times W}$, the full obstacle map $c_{gt} \in \mathbb{R}^{3 \times H \times W}$ and a binary mask $m \in \{0,1\}^{H \times W}$ where 0 indicates occluded regions and 1 denotes observed areas, we define the masked inputs as $s_m = m \odot s_{gt}$ and $c_m = m \odot c_{gt}$ through element-wise multiplication. In order to complement the map without prior knowledge, we employ a stochastic differential equation (SDE) (Song et al., 2021) formulation within the denoising diffusion probabilistic model (DDPM) (Ho et al., 2020) framework to reconstruct the complete visualized label maps from these corrupted observations. Specifically, in the case of an obstacle map, given that the

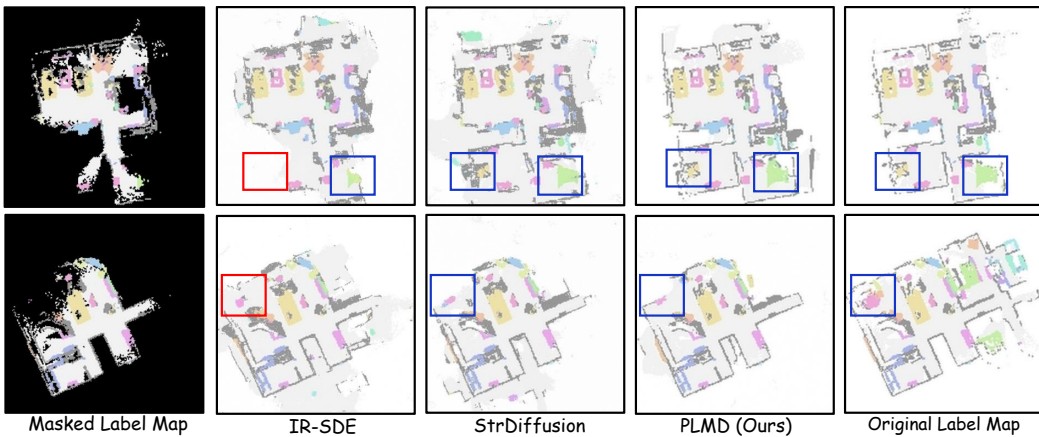

| Masked Label Map | IR-SDE | StrDiffusion | PLMD (Ours) | Original Label Map |

Figure 3: Label Map completion results. Label maps are derived from HM3D_v0.2 (val) and are not visible during PLMD training. The red boxes point out the missing parts of the restored visualized label maps.

initial obstacle map state $c_0 = c_{gt}$ and the final obstacle map state $c_T$ are set to the combination of the mask image $\mu_c = c_m$ with Gaussian noise $\mathcal{G}$, for any state $\tau \in [0, T]$, the diffusion process $\{c_\tau\}_{\tau=0}^T$ is defined via SDE as $dc = \theta_\tau(\mu_c - c)d\tau + \delta_\tau dw$, where $\theta_\tau$ and $\delta_\tau$ are time-dependent positive parameters, $\delta_\tau dw$ introduces stochasticity to the differential equation via a standard Wiener process $w$ (Song et al., 2020). The reverse denoising process operates under a time-reversed SDE:

$$dc = [\theta_\tau(\mu_c - c) - \delta_\tau^2 \nabla_c \log p_\tau(c)]d\tau + \delta_\tau d\hat{w}, \tag{1}$$

where $p_\tau(c)$ stands for the marginal probability density function of $c_\tau$ at time $t$ and $\hat{w}$ is a reverse-time Wiener process. The estimation of the score functions $\nabla_c \log p_\tau(c)$ is the key step in the reverse denoising process, which can be approximated by training a conditional time-dependent neural network $\mathcal{G}_\phi$ (Ho et al., 2020). We find the optimal reversed obstacle state $c_{\tau-1}^*$ from $c_\tau$ in time step $t-1$ by maximum likelihood learning. Given the state $c_\tau$ at time step $\tau$, we optimize $c_{\tau-1}$ by minimizing the negative log-likelihood:

$$c_{\tau-1}^* = \arg\min_{c_{\tau-1}}[-\log p(c_{\tau-1}|c_\tau, c_0)]. \tag{2}$$

Following the approach of DDPM, we input the state $c_\tau$, condition $\mu$, and time $\tau$ into the conditional time-dependent neural network $\mathcal{G}_\phi(c_\tau, \mu, \tau)$, which outputs pure noise. By optimizing $\mathcal{G}_\phi$ via the following objective:

$$\mathcal{L}_\alpha(\phi) = \sum_{\tau=1}^T \alpha_\tau \mathbb{E}[\|c_\tau - (dc_\tau)_{\mathcal{G}_\phi} - c_{\tau-1}^*\|_p], \tag{3}$$

where $\alpha_\tau$ is the positive weight, $(dc_\tau)_{\mathcal{G}_\phi}$ stands for the the reversetime SDE and $\|\cdot\|_p$ denote the $l_p$ norm, we first derive the final training target for the obstacle map network $\mathcal{G}_\phi$. Then we formulate the reverse process of the semantic map network $\tilde{\mathcal{G}}_\phi(s_\tau, c_{\tau-1}, \tau)$ by conditioning the semantic denoising on $c_{\tau-1}$, the refined obstacle map at timestep $\tau-1$, with the aim of jointly optimizing for the optimal reversed semantic state $s_{\tau-1}^*$:

$$s_{\tau-1}^* = \arg\min_{s_{\tau-1}}\left[-\log p(s_{\tau-1}|s_\tau, s_0, c_{\tau-1}, c_0)\right]. \tag{4}$$

We pre-train the obstacle map network $\mathcal{G}_\phi$ first to attain robust obstacle map priors before joint optimization with the semantic map network $\tilde{\mathcal{G}}_\phi(s_\tau, c_{\tau-1}, \tau)$. After pretraining the obstacle map network $\mathcal{G}_\phi$, we freeze it during semantic map network $\tilde{\mathcal{G}}_\phi$ training and cease further updates. $\tilde{\mathcal{G}}_\phi$ integrates obstacle map information via an obstacle-aware feature modulation network. Specifically, as shown in Fig. 2 (b), given the $k$-th feature map $f_\tau^k$, we adapt feature map $\hat{f}_\tau^k$ in $\tilde{\mathcal{G}}_\phi$ using a spatially-adaptive denormalization (Park et al., 2019) residual block:

$$\hat{f}_\tau^k = \mathbf{W}_\gamma^{(k)}(c_{\tau-1})f_\tau^k + \mathbf{b}_\beta^{(k)}(c_{\tau-1}), \tag{5}$$

where $\gamma$ and $\beta$ are modulation parameters, $\mathbf{W}_\gamma^{(k)}$ and $\mathbf{b}_\beta^{(k)}$ denote the mapping that converts the input $c_{\tau-1}$ to the scaled and biased values. *More details about $\hat{f}_\tau^k$ are provided in Appendix A.1.* On this basis, we turn to optimizing the semantic map network $\tilde{\mathcal{G}}_\phi(s_\tau, c_{\tau-1}, \tau)$ to estimate the optimal solution $s_{\tau-1}^*$ of the noise reduction process. To this end, the overall training objective to optimizing $\tilde{\mathcal{G}}$ is formulated:

$$\mathcal{L}_\zeta(\phi) = \sum_{\tau=1}^{T} \zeta_\tau \mathbb{E}[\|s_\tau - (ds_\tau)_{\tilde{\mathcal{G}}_\phi} s_{\tau-1} - s_{\tau-1}^*\|_p], \tag{6}$$

where $\zeta_\tau$ is the positive weight and $(ds_\tau)_{\tilde{\mathcal{G}}_\phi}$ denotes the estimated reverse noise by the noise network. These two pre-trained diffusion networks will be invoked simultaneously as the navigation task proceeds.

### 3.3 Navigation with Predicted Label Map

The trained PLMD serves as a plug-and-play auxiliary module, aiding any embodied navigation strategy in predicting unseen regions. We decouple PLMD-assisted navigation into Label Map Restored and Localization Strategy.

**Label Map Restored.** As illustrated in Fig. 2 (a), each robot processes its current RGB-D view $o_t$, sensor position $p_t$, and Goal **G** to construct semantic and obstacle map vectors. We use a fixed palette to populate the label indexes, resulting in a visualized semantic map $S_{vt}$ and a visualized obstacle map $C_{vt}$. For efficiency, these maps are cropped to retain only proximal valid (non-white) pixels around the robot, forming a local semantic map $s_t$ and masked version $s_t^m$, alongside an obstacle map $c_t$ and masked version $c_t^m$. A trained network $\mathcal{G}_\phi$ initializes the obstacle map $c_t^\tau$ with noise and iteratively denoises it via reverse SDE (Eq. 2) over $T$ steps to produce the final obstacle prediction $c_t^0$. Similarly, the semantic map $s_t^\tau$ is initialized with noise and denoised by $\tilde{\mathcal{G}}_\phi$, which integrates multi-scale obstacle priors from $\mathcal{G}_\phi$'s intermediate output $c_t^{\tau-1}$ using SPADE (Park et al., 2019) residual blocks to refine $s_t^\tau$. The outputs $c_t^0$ and $s_t^0$ are upscaled to match $C_{vt}$ and $S_{vt}$, then converted to predicted obstacle map vector $S_t^P$ of size $[2, H, W]$ (Channel $c_1$ and $c_2$) and predicted semantic map vector $C_t^P$ $[n, H, W]$ (Channel $s_1$ to $s_n$). We will concatenate $S_t^P$ and $C_t^P$ into predicted label map vector, which will then be merged into the new $M_t$ to replace the original map.

**Localization Strategy.** Before executing the navigation strategy, we directly search for goal pixel patches in $L_t^P$. To avoid interference from random discrete patches, for the set of goal pixel coordinates $X = \{x_1, x_2, \ldots, x_n\}$ in $L_t^P$, we use HDBSCAN (Campello et al., 2013) to extract and filter cluster labels $Z$: $Z = \text{HDBSCAN}(X, N)$, where $N$ is the parameter for the number of neighbors used to compute the core distance, which is empirically set to 5. We select the core of the densest cluster as the long-term goal and employ a local navigation strategy (e.g., Fast Marching Method (Sethian, 1999)) to reach it. Note that the cluster core may not exist or there may be multiple cores; we constrain this using a specific threshold. Specifically, cluster centers located within the $L_t^P$ are identified, and a composite score is calculated based on cluster density (weighted 50%), cluster size (weighted 40%) and distance to the starting point (weighted 10%). The center with the highest score is selected as the goal. If no valid clustering can be found within the region, the navigation strategy is executed. *We provide a summary of this process in the Appendix A.2.*

## 4 Experiments

### 4.1 Experimental Setup

**Datasets.** We evaluate PLN on ObjectNav (ON), Instance-ImageNav (IIN) and Multi-Robot ObjectNav (MRON). For ON, we conducted experiments on HM3D_v0.1 (Ramakrishnan et al., 2021), HM3D_v0.2 (Yadav et al., 2023b) and MP3D (Chang et al., 2017). For IIN, we compare with other methods on HM3D_v0.2. For MRON, we evaluate the performance of our model on HM3D_v0.2 and MP3D. *Detailed navigation benchmark settings can be found in the Appendix A.3.*

**Metrics.** To evaluate the navigation performance, we adopt two standard metrics (Yu et al., 2023b; Lei et al., 2024; Shen et al., 2024): 1) **SR**: the rate of successful events. 2) **SPL**: the path length-weighted

Table 1: Comparison of ObjectNav (ON), Instance-ImageNav (IIN) and Multi-Robot ObjectNav (MRON) on HM3D_v0.2, HM3D_v0.1, and MP3D. RL is reinforcement learning, SL denotes supervised learning and SSL is self-supervised learning. Bolding indicates the performance improvement over state-of-the-art navigation strategies.

| Method | Training | ON | | | | | | IIN | | MRON | | | |
|---|---|---|---|---|---|---|---|---|---|---|---|---|---|
| | | HM3D_v0.2 | | HM3D_v0.1 | | MP3D | | HM3D_v0.2 | | HM3D_v0.2 | | MP3D | |
| | | SR | SPL | SR | SPL | SR | SPL | SR | SPL | SR | SPL | SR | SPL |
| SemExp (Chaplot et al., 2020b) | RL | – | – | 0.379 | 0.188 | 0.360 | 0.144 | – | – | 0.612 | 0.327 | – | – |
| 3D-Aware (Zhang et al., 2023) | RL | – | – | – | – | 0.340 | 0.146 | – | – | – | – | – | – |
| OVRL-v2-IIN (Yadav et al., 2023a) | RL | – | – | – | – | – | – | 0.248 | 0.118 | – | – | – | – |
| IEVE (Lei et al., 2024) | RL | – | – | – | – | – | – | 0.702 | 0.252 | – | – | – | – |
| PLMD (Ours) | SSL+RL | – | – | **0.656** | **0.333** | **0.426** | 0.164 | **0.776** | **0.283** | – | – | – | – |
| PONI (Ramakrishnan et al., 2022) | SSL | – | – | – | – | 0.318 | 0.121 | – | – | – | – | – | – |
| SGM (Zhang et al., 2024) | SSL | – | – | 0.602 | 0.308 | 0.377 | 0.147 | – | – | – | – | – | – |
| T-Diff (Yu et al., 2024) | SSL | – | – | – | – | 0.396 | 0.152 | – | – | – | – | – | – |
| VLFM (Yokoyama et al., 2024) | × | – | – | 0.524 | 0.303 | 0.362 | 0.159 | – | – | – | – | – | – |
| OpenFMNav (Kuang et al., 2024) | × | – | – | 0.525 | 0.241 | 0.372 | 0.157 | – | – | – | – | – | – |
| SG-Nav (Yin et al., 2025a) | × | – | – | 0.540 | 0.249 | 0.402 | 0.160 | – | – | – | – | – | – |
| Mod-IIN (Krantz et al., 2023) | × | – | – | – | – | – | – | 0.561 | 0.233 | – | – | – | – |
| Co-NavGPT (Yu et al., 2023a) | × | 0.539 | 0.215 | – | – | – | – | – | – | 0.661 | 0.331 | – | – |
| MCoCoNav (Shen et al., 2024) | × | 0.634 | 0.297 | – | – | – | – | – | – | 0.716 | 0.387 | 0.568 | 0.334 |
| UniGoal (Yin et al., 2025b) | × | – | – | 0.545 | 0.251 | 0.410 | 0.164 | 0.602 | 0.237 | – | – | – | – |
| PLMD (Ours) | SSL | **0.665** | **0.302** | 0.618 | 0.304 | 0.412 | **0.167** | 0.642 | 0.244 | **0.762** | **0.406** | **0.591** | **0.382** |
| GT Label Maps | SSL/SSL+RL | 0.742 | 0.425 | 0.704 | 0.365 | - | - | 0.850 | 0.399 | 0.872 | 0.464 | 0.655 | 0.410 |

success rate, which measures the efficiency of the path length. In order to evaluate the quality of map completion, we introduce an additional metric: Peak Signal-to-Noise Ratio (**PSNR**), which is used to compare the low-level differences at pixel level between the generated image and ground-truth.

**Implementation details.** To train the PLMD, we collect obstacle and semantic maps of size $256 \times 256$ through the Habitat simulator (Savva et al., 2019). *Due to space constraints, training details are presented in Appendix A.5.* For the training set HM3D_v0.1 and MP3D, we uniformly utilize RedNet (Jiang et al., 2018) as the semantic segmentation tool to collect visualized map sequences with size $H = 480$, $W = 480$ from 25 starting timesteps in each scene and obtain the corresponding masks. The number of semantic Channels $n$ is set to 40. During training, we apply random rotations and flipping operations for data augmentation. The diffusion models are constructed by removing group normalization layers and self-attention layers from the U-Net in DDPM (Ho et al., 2020), and the Adam optimizer with $\beta_1 = 0.9$ and $\beta_2 = 0.99$ is employed. Based on experience, we set the timesteps of the diffusion model to $T = 100$. For RL, we use pre-trained SemExp (Chaplot et al., 2020b) weights. The global step is set to 25, 10 and 25 for the long-term policies of ON, IIN and MRON, respectively. All experiments are implemented under the PyTorch framework and run on 2 NVIDIA A40 GPUs. *Additionally, we provide detailed experiments setup (A.4), cross-dataset validation (A.6), and computational efficiency analyses (A.7, A.8).*

## 4.2 EVALUATION RESULTS

**Plug-in into Existing Methods.** As a plug-in, PLMD can be seamlessly combined with mainstream navigation strategies, effectively improving the original performance. For a fair comparison, we use OpenFMNav (Kuang et al., 2024) (ON on hm3d_v0.1), FBE (Yamauchi, 1997), and MCo-CoNav (Shen et al., 2024) (ON on hm3d_v0.2) as the navigation strategies for ON, IIN, and MRON, respectively, and PLMD is configured to activate after every 100 steps during navigation, then repeat every 50 steps thereafter. As shown in Table 1, for ON (hm3d_v0.1), PLMD outperforms the best method SGM (0.602) by 5.4% (row 5), and even without RL policy, it still achieves top-performing SR (row 15). Similarly, in IIN, PLMD combined with RL demonstrates more pronounced advantages (PLMD surpasses IEVE by 7%), showing greater improvement compared to RL-free conditions. This is because both ON and IIN tasks rely heavily on label map information for RL decision-making, where more complete label maps can fully unleash RL's planning potential. Furthermore, PLMD achieves state-of-the-art performance in both centralized (Yu et al., 2023a) and decentralized (Shen et al., 2024) MRON tasks that leverage LLMs. This indicates that accurately predicting unknown regions in semantic maps can effectively reduce the decision-making burden of modular navigation strategies. Additionally, we conduct experiments on HM3D_v0.1 (OpenFMNav (Kuang et al., 2024)), MP3D (MCoCoNav (Shen et al., 2024)) and HM3D_v0.2, comparing the navigational performance of PLMD prediction label maps with ground-truth label maps (ground-truth semantic and obstacle

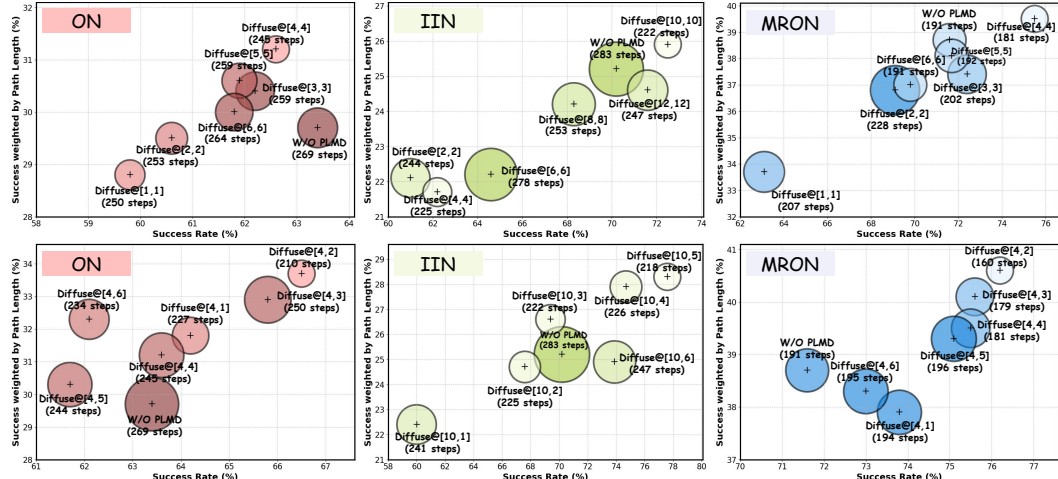

Figure 4: Visualization of the effect of PLMD execution frequency on navigation performance. Diffuse@$[x, y]$ indicates that PLMD execution starts from the $x$-th global step of navigation and repeats every $y$ global steps. The size of each point represents the average number of steps consumed in an episode of the navigation task.

distributions). The results further validate the importance of perfectly predicted label maps for the navigation strategy's performance.

**Comparison with Different Diffusion Methods.** To validate the superiority of PLMD in generating unknown regions of visualized label maps, we tested two diffusion model baselines, IR-SDE (Luo et al., 2023) and StrDiffusion (Liu et al., 2024), on the ON task in HM3D_v0.2, as shown in Table 2. Among these approaches, IR-SDE directly encodes semantic maps, while StrDiffusion leverages the progressive sparsity of structures to reduce semantic discrepancies. However, neither method considers the prior-driven role of obstacle maps in the denoising process for navigation tasks, and both operate solely at the

Table 2: Comparison of map unknown region generation performance of PLMD with different diffusion model baselines. All experiments were performed on the MRON task.

| Method | SR ↑ | SPL ↑ | PSNR ↑ |
|---|---|---|---|
| IR-SDE | 0.698 | 0.370 | 29.895 |
| StrDiffusion | 0.729 | 0.374 | 31.486 |
| **PLMD (Ours)** | **0.762** | **0.406** | **34.284** |

pixel level (obstacles are not incorporated into the label map). As shown in Fig. 3, compared to IR-SDE, StrDiffusion reduces artifacts in the generated prediction maps. Nevertheless, it remains constrained by semantic maps. Specifically, it struggles to predict obstacles in unknown regions (e.g., missing obstacles in StrDiffusion's predictions in Fig. 3). Our analysis suggests due to the fact that sparse obstacle structures are inherently more difficult to learn than dense semantic features. In contrast, PLMD replaces semantic maps with label maps, effectively capturing the contextual relationship between obstacles and semantic features. Overall, PLMD leads to more efficient navigation performance, as it explicitly models the obstacle structure a priori, rather than relying solely on pixel-level semantic predictions.

**Discussion of PLMD Execution Frequency.** The execution frequency of PLMD is fixed as follows: it starts at the 100-th navigation step and is repeated every 50 steps thereafter. This setting remains effective across multiple tasks without requiring dynamic adjustment. To validate this, we conducted extensive evaluations of ON, IIN, and MRON using IEVE (IIN) and MCoCoNav (ON, MRON) as navigation strategies on HM3D_v0.2. Based on the results shown in Fig. 4, we offer the following two key observations: ❶ **Initiating PLMD at the 100-th step (corresponding to 4 global steps for ON/MRON or 10 global steps for IIN) is most appropriate.** In the IIN and MRON tasks, navigation performance generally improves as the starting step of PLMD increases, with the configuration Diffuse@$[10, 10]$ and Diffuse@$[4, 4]$ achieving peak performance (2.3% / 3.9% improvement in SR, 0.5% / 0.8% improvement in SPL, and 61 / 10 decrease in the number of navigation steps compared to the baseline without PLMD), and navigation performance gradually decreases as the number of starting step continues to increase. For the ON task, although the SR

Table 3: Comparison of cluster weight distribution selections. The values in Weight Distributions correspond to cluster density, cluster size, and distance to the starting point.

| Weight Distributions | HM3D SR↑ | HM3D SPL↑ | MP3D SR↑ | MP3D SPL↑ |
|---|---|---|---|---|
| 0.5 / 0.4 / 0.1 | **0.762** | **0.406** | **0.591** | **0.382** |
| 0.7 / 0.2 / 0.1 | 0.758 | 0.399 | 0.588 | 0.382 |
| 0.2 / 0.7 / 0.1 | 0.743 | 0.392 | 0.562 | 0.363 |
| 0.1 / 0.2 / 0.7 | 0.740 | 0.385 | 0.540 | 0.366 |
| 0.33 / 0.33 / 0.34 | 0.741 | 0.385 | 0.577 | 0.375 |

under the Diffuse@$[4, 4]$ configuration is slightly lower than that of the baseline without PLMD, it still maintains a high SPL, and significantly reduces the average number of navigation steps by approximately 9%, indicating that PLMD enhances overall efficiency by reducing unnecessary exploration. ❷ **Starting PLMD at the 100th navigation step and repeating every 50 steps yields optimal efficiency improvements across different task types.** Specifically, for ON and MRON, the Diffuse@$[4, 2]$ configuration achieves the highest SR (66.5% / 76.2%) and SPL (30.2% / 40.6%), along with the lowest average number of navigation steps (210 / 160). For IIN, the best performance is achieved under the Diffuse@$[10, 5]$ configuration. These results demonstrate that PLMD must be synchronized with the robot's exploration progress: prediction should begin after sufficient environmental data has been collected (after 100 steps), and the map should be refreshed at regular intervals (every 50 steps). Overall, for various navigation tasks, PLMD does not require dynamic adjustment of its execution frequency.

**Analysis of Cluster Weight Distributions Selection.** We evaluated the performance of five representative weight distributions (cluster density/cluster size/distance to the starting point) in Table 3: 1) our initial design (0.5 / 0.4 / 0.1), 2) density-dominant (0.7 / 0.2 / 0.1), 3) size-dominant (0.2 / 0.7 / 0.1), 4) distance-dominant (0.1 / 0.2 / 0.7), and 5) uniform distribution (0.33 / 0.33 / 0.34). We conducted MRON experiments on HM3D_v0.2 and MP3D validation sets, with evaluations for the subsets provided in Appendix A.12. Results indicate that the initial design weighting (0.5 / 0.4 / 0.1) consistently achieved optimal performance. In contrast, weighting schemes overly biased towards single factors failed to comprehensively outperform the original design, while the uniform weight distribution (0.33 / 0.33 / 0.34) also yielded slightly inferior results. We observe that cluster density and cluster size contribute more directly to navigation success than distance to the starting point.

**Evaluations of Open-Vocabulary Goal.** To further validate the PLMD's generalisation capability in open-vocabulary scenarios, we extended the evaluation process by retraining the PLMD using the Grounded SAM (Ren et al., 2024) as open-vocabulary segmentation method. We then conducted MRON experiments on the HM3D_v0.2 validation set for unseen categories (lamp, toy car, microwave), with evaluations for the subset provided in Appendix A.13. As shown in Table 4, the Grounded SAM-based PLMD (PLMD[†]) outperformed the original PLMD and other baseline methods in both

Table 4: Evaluations of open-vocabulary goal. We additionally select objects (lamp, toy car, microwave) not included in the standard HM3D and MP3D validation set for testing.

| Method | SR↑ | SPL↑ | Total time (s) |
|---|---|---|---|
| Multi-SemExp | 0.285 | 0.206 | 263.5 |
| MCoCoNav | 0.327 | 0.242 | 1063.6 |
| PLMD (Ours) | 0.323 | 0.225 | 1195.7 |
| PLMD[†] (Ours) | **0.354** | **0.268** | 1535.6 |

SR and SPL, achieving 0.354 and 0.268 respectively. This demonstrates that our proposed obstacle-aware diffusion architecture can be effectively transferred to open-vocabulary goal navigation tasks. However, we also note that the introduction of the Grounded SAM module substantially increases the computational overhead of single-step reasoning, leading to an overall increase in navigation time for PLMD[†], which points the way for future efficiency optimisations.

**Navigation Visualization.** Fig. 5 illustrates the visualization of the PLMD-assisted MRON navigation process for searching the goal 'Chair'. Notably, this visualization was conducted on the HM3D_v0.2 validation set, while our PLMD was trained on HM3D_v0.1, with no prior knowledge of the scene layout. Despite this, PLMD effectively expands the label map, enabling the navigation strategy to infer the goal's location more accurately. *Due to space constraints, the navigation visualizations for ON and IIN are given in Appendix A.10.*

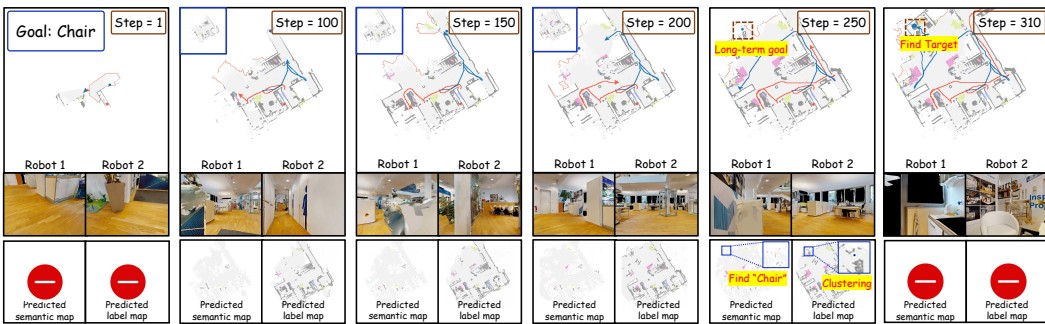

Figure 5: Visualization of the PLMD navigation process (MRON). The upper column includes the navigation goal, the current navigation timestep, the RGB view and the semantic map constructed by the robots at each navigation timestep. The small blue boxes represent the semantic map after removing the robot, navigation trajectory, and long-term target points. The lower column displays the predicted visualized semantic maps and label maps. Best viewed when zoomed in.

Table 5: Ablations for PLMD-assisted navigation tasks.

| Condition | ON | | | IIN | | | MRON | | |
|---|---|---|---|---|---|---|---|---|---|
| | SR ↑ | SPL ↑ | PSNR ↑ | SR ↑ | SPL ↑ | PSNR ↑ | SR ↑ | SPL ↑ | PSNR ↑ |
| (Diffusion) w/o $\mathcal{G}_\phi$ | 0.636 | 0.285 | 30.437 | 0.730 | 0.264 | 30.437 | 0.714 | 0.358 | 30.437 |
| w/o Obstacle Map | 0.626 | 0.284 | 34.284 | 0.727 | 0.271 | 34.284 | 0.717 | 0.363 | 34.284 |
| w/o Clustering | 0.657 | 0.303 | 34.284 | 0.757 | 0.277 | 34.284 | 0.748 | 0.395 | 34.284 |
| Image Replacement | 0.640 | 0.273 | 34.284 | – | – | – | 0.731 | 0.394 | 34.284 |
| **PLMD** | **0.665** | **0.302** | 34.284 | **0.776** | **0.283** | 34.284 | **0.762** | **0.406** | 34.284 |

**Ablation studies.** Considering that our method contains both Map Completion and Navigation phases, we perform the following ablations on HM3D_v0.2: 1) The impact of network $\mathcal{G}_\phi$. 2) The effect of whether or not to use obstacle maps. 3) The impact of clustering. 4) Replacement of visualized semantic maps in the navigation strategy (not applicable for RL). As shown in Table 5, the results reveal that the network $\mathcal{G}_\phi$ and the obstacle map play a decisive role of ON, IIN and MRON tasks: removing $\mathcal{G}_\phi$ (i.e., not using the obstacle map as a priori) reduces SR by 2.4% (ON), 4.6% (IIN) and 4.8% (MRON), and the quality of label maps gets worse (PSNR decreases by 3.847). While eliminating the obstacle map leads to more significant decreases of 3.9% (ON) and 4.9% (IIN) for ON and IIN, with MRON decreasing by 4.5%. Notably, the clustering mechanism for Localization Strategy is particularly crucial for long-term planning in IIN, with its absence causing a 1.9% SR decline, while it has a smaller effect on ON (-0.8%) and MRON (-1.4%). Furthermore, replacing the visual semantic map with diffusion-generated outputs results in performance drops (ON -1.5%, MRON -3.1%), yet still demonstrates PLMD's plug-and-play compatibility across tasks. These findings demonstrate the synergistic design of obstacle map-driven semantic map diffusion model with vectorized label map representations that can effectively meet universal embodied navigation needs, particularly excelling under unseen scenarios.

## 5 CONCLUSION

In this work, we present the Plug-and-Play Label Map Diffusion (PLMD) approach, which effectively mitigates semantic inconsistencies in BEV maps generation by integrating obstacle prior information into the semantic denoising process. PLMD can be seamlessly plugged into map-dependent GON strategies while augmenting multiple navigation strategies by generating label-level complete maps and pixel-level localization. Furthermore, it outperforms the original navigation strategy across multiple datasets without requiring retraining. We believe that PLMD can facilitate the advancement toward larger-scale universal embodied navigation.

## 6 ETHICS STATEMENT

This research adheres to the ICLR Code of Ethics. Our work focuses on embodied Goal-Oriented Navigation within simulated environments (HM3D and MP3D) and does not involve human subjects, personally identifiable information, or sensitive data. All datasets used are publicly available and widely adopted in the embodied AI community. We have carefully considered the potential societal and environmental impacts of this research. The method does not directly pose risks to safety, security, or privacy. We encourage responsible deployment that prioritizes human well-being and avoids applications that could lead to surveillance misuse or inequitable impacts. All experiments are reported transparently, with sufficient detail to ensure reproducibility. No part of this work involved deception, fabrication, or falsification of data.

## 7 REPRODUCIBILITY STATEMENT

We have made every effort to ensure the reproducibility of our results. The main paper provides a detailed description of the proposed PLMD framework, including its architecture, training objectives, and integration into navigation strategies (Section 3). Experimental settings, datasets, and evaluation metrics are described in Section 4, with additional implementation details, benchmark configurations, and algorithmic workflows provided in the Appendix (Sections A.1-A.6). To facilitate verification, we release anonymized source code, configuration files, and demonstration videos as supplementary material. All datasets used in our experiments (HM3D and MP3D) are publicly available, and we describe preprocessing steps in the Appendix. Together, these resources are intended to ensure that our results can be independently reproduced and extended by the community.

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

## A   APPENDIX

### A.1   MORE DETAILS ABOUT $\hat{f}_\tau^k$

Given the feature map $\hat{f}_\tau^k$ of the upper layer of the denoising network, the mask is first projected into the embedding space and then convolved to generate the modulation parameters $\gamma$ and $\beta$. Unlike conditional normalization methods, $\gamma$ and $\beta$ are not vectors but tensors with spatial dimensions. The generated $\gamma$ and $\beta$ are multiplied and added to the normalized activation elements. We use the $k$-th layer feature mapping $f_\tau^k \in \mathbb{R}^{C^k \times H^k \times W^k}$ for obstacle map driving:

$$
\begin{aligned}
\hat{f}_\tau^k &= \mathbf{W}_\gamma^{(k)}(c_{\tau-1})f_\tau^k + \mathbf{b}_\beta^{(k)}(c_{\tau-1}) \\
&= \mathbf{W}_\gamma^{(k)}(c_{\tau-1})\frac{h_\tau^k - \mu_t^k}{\sigma_\tau^k} + \mathbf{b}_\beta^{(k)}(c_{\tau-1}),
\end{aligned}
\tag{7}
$$

where $h_\tau^k$ is the activation at the site before normalization and $\mu_\tau^k$ and $\sigma_\tau^k$ are are the statistical mean and variance of the pixels across different channels $\mathcal{C}$:

$$
\begin{aligned}
\mu_\tau^k(h^k, w^k) &= \frac{1}{\mathcal{C}^k} \sum_{c^k=1}^{\mathcal{C}^k} h_\tau^k(h^k, w^k, c^k), \\
\sigma_\tau^k(h^k, w^k) &= \sqrt{\frac{1}{\mathcal{C}^k} \sum_{c^k=1}^{\mathcal{C}^k} (h_\tau^k(h^k, w^k, c^k) - \mu_\tau^k(h^k, w^k))^2}, \\
h^k &= 1, 2, ..., H^k, w^k = 1, 2, ..., W^k,
\end{aligned}
\tag{8}
$$

where $\sigma_\tau^k(h^k, w^k)$ and $\mu_\tau^k(h^k, w^k)$ are the statistical mean and variance of the pixels across different channels at the position $(h^k, w^k)$.

---

**Algorithm 1** Localization Strategy

---

**Require:**
    Navigation timestep $t$, Predicted label map $L_t^P$, Start coordinates **start**
**Ensure:**
    Navigation goal point **goal**
 1: **while** navigation is not completed **do**
 2:     Acquire current visualized predicted label map $L_t^P$
 3:     Cluster analysis: $\{\mathbf{centers}\} \leftarrow \text{HDBSCAN}(L_t^P)$
 4:     **if** valid cluster centers exist **then**
 5:        **if** candidate set is not empty **then**
 6:           Compute composite score: $score = 0.5\frac{1}{density} + 0.4size + 0.1\frac{1}{distance}$
 7:           $\mathbf{goal} \leftarrow \arg\max(score)$
 8:        **end if**
 9:     **else**
10:        Executing the navigation strategy
11:     **end if**
12:     Navigate to **goal**
13:     $t \leftarrow t + 1$
14: **end while**

---

### A.2   MORE DETAILS ABOUT LOCALIZATION STRATEGY

We further provide the workflow of Section 3.3 Localization Strategy. Based on the PLMD, the Positioning Strategy is summarized in Algorithm 1.

### A.3   NAVIGATION BENCHMARK SETTINGS

**ObjectNav (ON).** For the ON task, we use the following setup: The robot has a height of 0.88 meters and a radius of 0.18 meters. It receives a $640 \times 480$ RGB-D egocentric view from a camera positioned

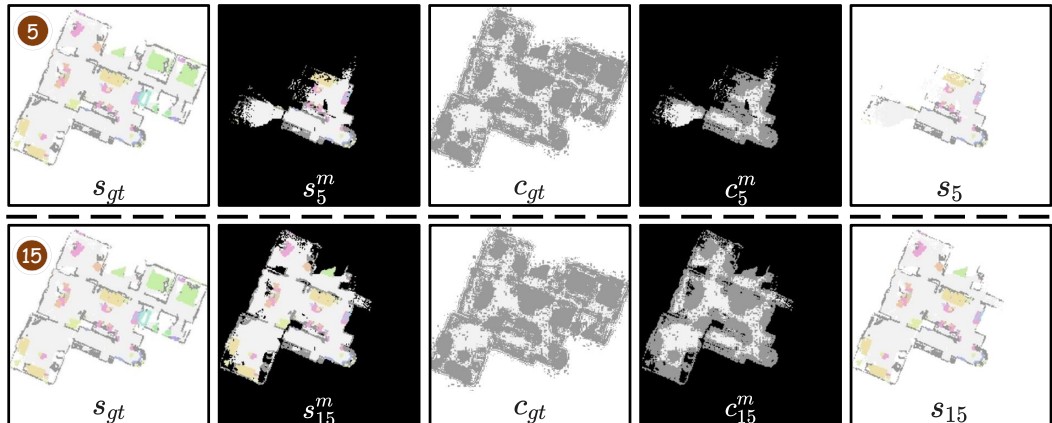

Figure 6: Visualization of label map observation pair $\{(s_{gt}, s_5^m, c_{gt}, c_5^m)\}$ and $\{(s_{gt}, s_{15}^m, c_{gt}, c_{15}^m)\}$.

0.88 meters above the ground with a 79° horizontal field of view (HFoV). The action space consists of six actions: move forward, turn left, turn right, look up, look down, and stop. The movement step size is 0.25 meters, and each rotation action turns the robot by 30°. In the MP3D and HM3D datasets, the robot receives its GPS position at each time step. The robot is initialized at a random position in the scene and receives the goal object category. At each time step, the robot observes the environment and takes an action. The stop action is used when the robot is close to the goal object. An episode is considered successful if the robot takes the stop action within a distance of less than 0.2 meters from the goal. The maximum number of time steps per episode is 500; if steps exceed 500, the task fails. For HM3D_v0.2, the validation split includes 1,000 episodes, spanning 36 scenes and 6 object categories. For the training and validation sets of HM3D_v0.1, we choose 80 train / 20 val scenes, 6 goal categories and 2,000 training and validation episodes. For the training and validation sets of MP3D, we utilize 56 train / 11 val scenes, with 2,195 training and validation episodes containing 21 goal object categories.

**Instance-ImageNav (IIN).** For the IIN task, we make the following changes while keeping the ON settings: The linear speed is capped at a maximum of $0.35\text{m}/frame$ and angular velocity at $60°/frame$ in the action space. Success is True if the robot calls velocity stop action within 1.0m Euclidean distance of the goal object and the object is oracle-visible by turning or looking up and down.

**Multi-Robot ObjectNav (MRON).** For the MRON task, we make the following changes while keeping the ON settings: The stop action is triggered when one of the robots $Robot^i$ approaches the goal object. An episode is considered successful if the distance between robot $Robot^i$ and the goal is less than 0.1m and robot $Robot^i$ executes the stop action.

A.4    EXPERIMENTS SETUP

We employ SR and SPL to evaluate the embodied navigation performance and assess the visualized label map generation accuracy by PSNR metric. Note that, since the navigation performance of multiple repeated experiments does not show significant differences, error bars are not reported.

**SR (Success Rate).** SR measures the success rate of the robot in successfully finding the goal object. It is defined as $SR = \frac{1}{N} \sum_{i=1}^{N} S_i$, where $N$ is the total number of validation episodes and $S_i$ is an indicator that representing whether the i-th episode is successful or not.

**SPL ((Success weighted by Path Length).** SPL measures the success of the robot weighted by the efficiency of the path taken. It is defined as $SPL = \frac{1}{N} \sum_{i=1}^{N} S_i \cdot \frac{l_i}{\max(L_i, l_i)}$, where $N$ is the total number of validation episodes, $S_i$ is an indicator variable that equals 1 if the i-th episode is successful and 0 otherwise, $l_i$ is the length of the path actually taken by the robot in the i-th episode, and $L_i$ is the length of the shortest possible path to the goal for that episode.

**PSNR (Peak Signal-to-Noise Ratio).** PSNR is a metric used to measure the quality of a reconstructed image. It quantifies the ratio between the maximum possible power of a signal and the power of

Table 6: PLMD evaluations on MP3D.

| Training dataset | MRON SR ↑ | MRON SPL ↑ | PSNR ↑ |
|---|---|---|---|
| HM3D_v0.1 | 0.762 | 0.406 | 34.284 |
| MP3D | 0.755 | 0.395 | 33.411 |

distorting noise that affects the fidelity of the representation. PSNR is defined as:

$$\text{PSNR} = 20 \cdot \log_{10}\left(\frac{\text{MAX}_I}{\sqrt{\text{MSE}}}\right), \tag{9}$$

where $\text{MAX}_I$ is the maximum possible pixel value of the image (e.g., 255 for 8-bit grayscale images), and MSE is the mean squared error between the original and the reconstructed image, given by:

$$\text{MSE} = \frac{1}{mn}\sum_{i=0}^{m-1}\sum_{j=0}^{n-1}(I(i,j) - \hat{I}(i,j))^2, \tag{10}$$

where $I$ is the original image, $\hat{I}$ is the reconstructed image, and $m \times n$ is the size of the image. Higher PSNR values indicate better reconstruction quality, with less distortion relative to the original signal.

## A.5 MORE DETAILS ABOUT PLMD TRAINING SET $\mathcal{T}$ AND VALIDATION SET $\mathcal{V}$ ON HM3D_v0.1

To collect sufficient training data, we extensively explore all navigable environments in the HM3D_v0.1 training dataset using the Frontier-Based Exploration (FBE) strategy, aiming to achieve full scene coverage. This process yields 40,000 label map observation pairs $\mathcal{O}_{label}$. We then apply cropping and rotation to these pairs, and further filter out samples with insufficient semantic or obstacle pixels (threshold set at 100 non-white pixels), resulting in a final dataset of 238,800 processed observation pairs. The dataset is randomly split into a training set and a validation set at an 8:2 ratio, yielding the final training set $\mathcal{T}$ containing 191,040 pairs and the validation set $\mathcal{V}$ containing 47,760 pairs with corresponding scenes. We use only scenarios from the validation set as val scenes during navigation in HM3d_v0.1. Fig. 6 illustrates two example pairs from the dataset, corresponding to observations at the 5-th and 15-th global steps in navigation. Notably, unlike previous work, we do not use ground-truth semantics as the training dataset but instead generate training data from label maps collected during robot interactions, as directly acquiring semantic information from ground-truth is impractical in real-world scenarios. The ground-truth semantics here refers to ground-truth semantics in the navigation scene are obtained directly without using a semantic segmentation model. All semantics used in the map during PLMD training are obtained by the semantic segmentation model.

## A.6 PLMD EVALUATIONS ON MP3D

To quantify PLMD's robustness, we further trained PLMD on the MP3D dataset under identical configurations as in Section 4.1 and Section A.5 (we omitted HM3D_v0.2 due to its scene similarity with HM3D_v0.1). For MP3D, we collected 471,900 processed observation pairs in the training scenes, comprising 377,520 training pairs and 94,380 validation pairs. The results in Table 6 demonstrate that PLMD (MP3D) achieves comparable performance to PLMD (HM3D_v0.1), confirming its adaptability to unseen HM3D_v0.2 environments. Notably, PLMD (MP3D) exhibits marginally lower metrics than PLMD (HM3D_v0.1), attributable to MP3D's inferior 3D scan quality relative to HM3D_v0.1. Additionally, our cross-dataset evaluation on both HM3D_v0.2 and MP3D (as shown in Table 1) verifies PLMD's robust out-of-distribution performance across diverse environments.

## A.7 COMPUTATIONAL EFFICIENCY

As shown in Table 7, to quantify the computational complexity of PLMD, we report a comparison of Floating Point Operations (FLOPs) between PLMD and other navigation frameworks (higher FLOPs values indicate higher computational complexity). It is important to note that PLMD begins working after 100 steps of navigation. For each time step, it iterates 100 steps to generate the label

Table 7: FLOPs of SemExp (Chaplot et al., 2020b), PONI (Ramakrishnan et al., 2022), OpenFM-Nav (Kuang et al., 2024) and our PLMD.

| Method | SemExp | PONI | OpenFMNav (Qwen2.5-VL 7B) | PLMD (ours) |
|---|---|---|---|---|
| FLOPs (G) | 3.1 | 46.6 | 276.9 | 34.5 |

Table 8: The proportion of diffusion model inference time in the total navigation time. 'PLMD inference time' represent average PLMD inference time per episode and 'Total time' is the average total time per episode. In the 'Total time' column, rows without '(with LLMs)' denote reinforcement learning-based approaches.

| Task | PLMD inference time (s) | Total time (s) |
|---|---|---|
| ON | 139.1 (15.95%) | 872.1 (with LLMs) |
| ON | 52.9 (20.15%) | 262.5 |
| MRON | 120.5 (9.66%) | 1247.0 (with LLMs) |
| MRON | 79.7 (19.67%) | 405.2 |
| IIN | 68.1 (20.80%) | 327.4 |

map vector and repeats this process every 50 steps during navigation. Therefore, for navigation tasks utilizing PLMD (such as SemExp), we calculate the computational complexity of 100 iterations and average it over every 50 steps after 100 navigation steps. The results show that although PLMD (map resolution $256 \times 256$) requires multiple iterations, its computational overhead is superior to PONI (map resolution $480 \times 480$) or methods based on large language models (LLMs). Therefore, the computational complexity of PLMD is considered acceptable.

## A.8 COMPUTATIONAL TIME TRADE-OFFS

PLMD operates asynchronously with navigation, activating only at critical intervals (every 50 steps after the initial 100 navigation steps, as shown in Fig. 4), thereby achieving benefits without significantly affecting real-time navigation performance. We report the proportion of time consumed by diffusion model inference during navigation relative to the total navigation time, as shown in Table 8 (values in parentheses indicate the proportion of time spent on PLMD). We observe that PLMD accounts for 10%-20% of the total inference time. When we removed the obstacle map network, navigation performance decreased significantly, while the overall inference time changed little. The results indicate that although PLMD requires time for inference, it is acceptable compared to the total navigation time.

## A.9 NETWORK DESIGN CHOICES ABLATIONS

Semantic maps capture object categories and contextual relationships, while obstacle maps represent geometric and navigable area constraints. By decoupling these two maps, we enable each diffusion model to specialize in its respective domain, thereby achieving the goal of synergistic optimization between obstacle map priors and semantic maps. Our ablation studies (Table 5) demonstrate that removing either component significantly degrades performance (e.g., MRON's SR drops by 4.8% when omitting $\mathcal{G}_\phi$), validating our design choice and underscoring the importance of separate map processing.

Through comparative ablation experiments (see Table 9), we evaluated three alternatives as shown in Fig. 7: (1) CNN fusion module with two sequential $3 \times 3$ convolutional layers and residual connections, (2) an attention mechanism with Query/Key/Value generated via three independent $1 \times 1$ convolutions, and (3) our SPADE implementation. We found that compared to the attention mechanism, SPADE's lightweight affine transformation (Eq. 5) demonstrates higher time efficiency during the iterative denoising process (100 steps), increasing navigation time overhead by only 6.1% compared to CNN fusion, while the cross-attention mechanism increases it by 14.3%. Although we acknowledge that CNN fusion may be simpler, our ablation experiments show that it reduces overall navigation SR, indicating the rationality of choosing SPADE.

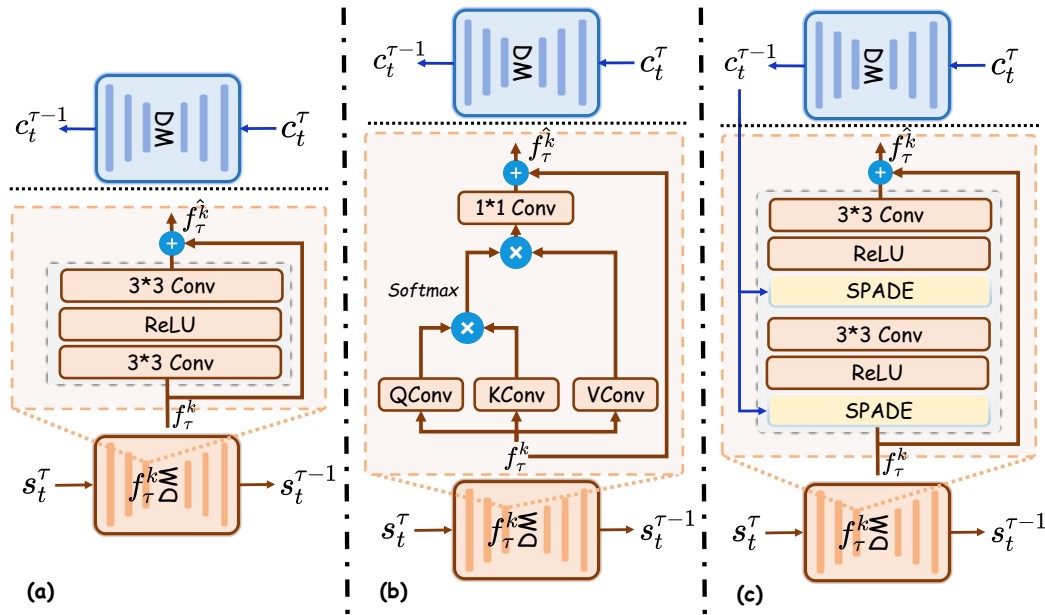

Figure 7: Three distinct network design choices: **(a)** CNN Fusion module; **(b)** Attention Fusion module; **(c)** SPADE module employed by PLMD.

Table 9: Ablations for network design choices.

| Component | IIN SR ↑ | IIN SPL ↑ | PLMD inference time (s) | Total time (s) |
|---|---|---|---|---|
| CNN Fusion | 0.728 | 0.247 | 55.9 (18.12%) | 308.5 |
| Attention Fusion | 0.716 | 0.249 | 71.7 (20.46%) | 350.5 |
| SPADE | 0.776 | 0.283 | 68.1 (20.80%) | 327.4 |

## A.10 MORE NAVIGATION VISUALIZATIONS

Fig. 8 provides a visualization of the navigation process for the ON task in searching for the goal 'Bed'. When the step count is below 100, due to limited observations, there is insufficient semantic and obstacle information for prediction, so only the original navigation strategy is executed. When the step count reaches 100, we integrate the label map information from the PLMD-generated map (e.g., object types represented by pixels around frontiers) into the OpenFMNav navigation strategy for long-term goal selection. At step 200, the ReasonLLM module in OpenFMNav utilizes the label map predicted by PLMD to infer the frontier most likely to contain the goal (Frontier 3), followed by further exploration. Ultimately, at navigation step 239, the robot successfully locates the goal.

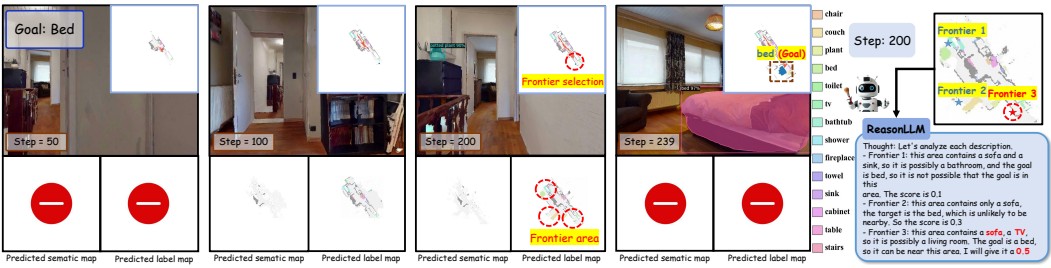

Figure 8: Visualization of the PLMD navigation process with OpenFMNav (Kuang et al., 2024) (ON). The upper column includes the navigation goal, the current navigation timestep, the RGB view and the semantic map constructed by the robots at each navigation timestep. The lower column displays the predicted visualized semantic maps and label maps. Best viewed when zoomed in.

Figure 9: Visualization of the PLMD navigation process with IEVE (Lei et al., 2024) (IIN). The upper column includes the navigation goal (instance image), the current navigation timestep, the RGB view and the semantic map constructed by the robots at each navigation timestep. The lower column displays the predicted visualized semantic maps and label maps. Best viewed when zoomed in.

Table 11: Comparison between A40 and Edge Computing Devices in Instance-ImageNav (IIN) tasks.

| Device | IIN SR ↑ | IIN SPL ↑ | PLMD inference time (s) | Total time (s) |
|---|---|---|---|---|
| A40 | 0.776 | 0.283 | 68.1 (20.80%) | 327.4 |
| Jetson AGX Orin | 0.747 | 0.264 | 177.5 (20.81%) | 852.7 |

Additionally, as shown in Fig. 9, for the IIN task, PLMD accurately generates predictive label maps based on the current semantic map and transforms the pixel-level maps into vector-level maps, which effectively guides the RL strategy to direct the robot to the goal object. These confirm the analysis presented in the main body: PLMD effectively extends the unknown map region by generating predicted label map, thereby assisting the navigation strategy in inferring the goal's location.

In addition, we provide video demos of ON, IIN, and MRON that show a more intuitive PLMD-assisted navigation process. Please refer to the MP4 file in the supplements zip archive. Fixed color palettes are also provided in the Supplementary Material.

## A.11 DISCUSSION OF REAL-TIME NAVIGATION REQUIREMENTS FOR ROBOTS

We deployed PLMD on the NVIDIA Jetson AGX Orin 64GB edge device, demonstrating its IIN performance in comparison to the A40 GPU. We also reported the time consumption of deploying the two devices. As shown in Table 11, the results show that when deployed on Jetson AGX Orin, despite computational limitations, we were able to maintain a success rate of 96% (0.747 SR) for A40, demonstrating the feasibility of PLMD on resource-constrained devices (The percentage of time cost occupied by PLMD did NOT show a significant increase).

Table 10: Evaluations of open-vocabulary goal on validation subset.

| Method | SR↑ | SPL↑ | Total time (s) |
|---|---|---|---|
| Multi-SemExp | 0.317 | 0.222 | 246.8 |
| MCoCoNav | 0.399 | 0.265 | 1007.1 |
| PLMD (Ours) | 0.363 | 0.232 | 1169.2 |
| PLMD[†] (Ours) | **0.446** | **0.274** | 1540.2 |

We propose several optimization measures to address the latency issues in diffusion model inference: (1) The diffusion process is activated periodically (every 50 steps) rather than continuously, significantly reducing computational load; (2) Our plug-and-play design allows for reduced prediction quality by decreasing diffusion steps or resolution when needed, thereby improving speed.

## A.12 ANALYSIS OF CLUSTER WEIGHT DISTRIBUTIONS SELECTION ON SUBSET

We randomly selected 200 episodes each from HM3D_v0.2 and MP3D as the MRON validation subset. In Table 12, we provide the comparison of cluster weight distribution selections on this subset.

Table 12: Comparison of cluster weight distribution selections on the subset. The values in Weight Distributions correspond to cluster density, cluster size, and distance to the starting point.

| Weight Distributions | HM3D SR↑ | HM3D SPL↑ | MP3D SR↑ | MP3D SPL↑ |
|---|---|---|---|---|
| 0.5 / 0.4 / 0.1 | **0.726** | **0.401** | **0.588** | **0.382** |
| 0.7 / 0.2 / 0.1 | 0.721 | 0.395 | 0.585 | 0.382 |
| 0.2 / 0.7 / 0.1 | 0.711 | 0.398 | 0.552 | 0.344 |
| 0.1 / 0.2 / 0.7 | 0.707 | 0.385 | 0.537 | 0.351 |
| 0.33 / 0.33 / 0.34 | 0.715 | 0.384 | 0.572 | 0.369 |

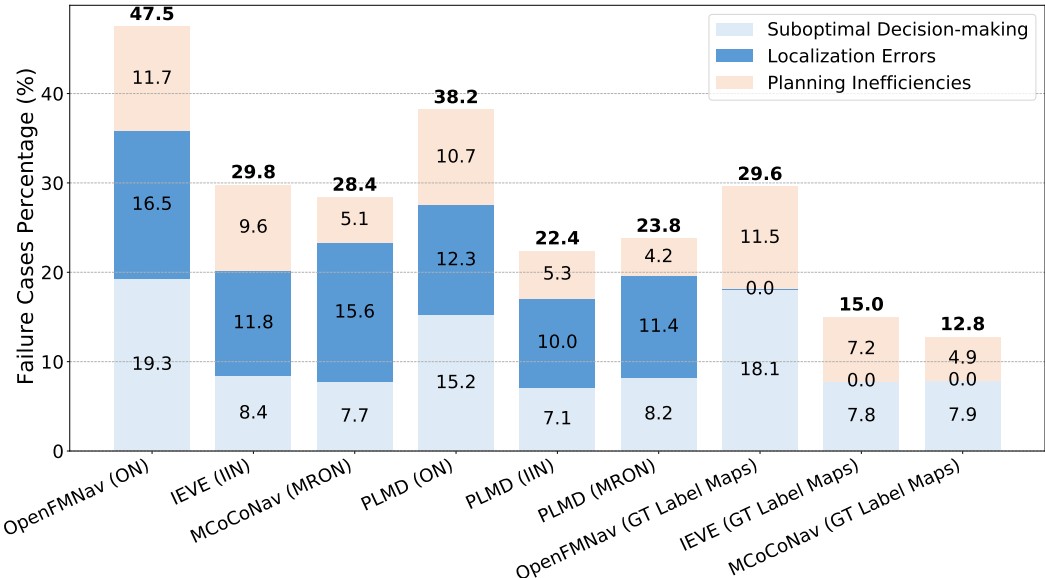

Figure 10: Percentage of failure cases in different baselines.

## A.13 EVALUATIONS OF OPEN-VOCABULARY GOAL ON SUBSET

Following the setup in A.12, we provide Open-Vocabulary Goal navigation performance on subset in Table 10.

## A.14 DIFFUSION STEP ABLATIONS

We conducted additional ablation studies on NVIDIA Jetson AGX Orin with reduced diffusion steps ($T = 25, 50, 75$) and higher diffusion steps ($T = 150, 200$). The results in Table 13 demonstrate a trade-off between navigation performance and computational cost: while reducing $T$ from 200 to 100 decreases IIN SR by only 0.5%, it achieves a 31.4% reduction in total time (from 1242.3s to 852.7s); however, further reducing $T$ (from 100 to 25) leads to a more significant 6.7% drop in IIN SR despite greater time savings. The optimal balance occurs at $T = 100$, where navigation performance peaks with acceptable time consumption. Additionally, as shown in Table 14, experiments on A40 demonstrate further performance improvements compared to results on NVIDIA Jetson AGX Orin.

## A.15 FAILURE CASES STUDY

We analyzed all failure instances across episodes and classified them into *Suboptimal Decision-making*, *Localization Errors*, and *Planning Inefficiencies*. *Suboptimal Decision-making* happens when the robot reaches the maximum number of navigation steps without finding the goal. *Localization Errors* occur when the robot makes an error in localization. *Planning Inefficiencies* happen when

Table 13: Ablation for diffusion step $T$ on NVIDIA Jetson AGX Orin.

| T | IIN SR↑ | IIN SPL↑ | PLMD inference time (s) | Total time (s) |
|---|---------|----------|------------------------|----------------|
| 200 | 0.752 | 0.270 | 261.8 | 1242.3 |
| 150 | 0.750 | 0.268 | 222.2 | 1085.7 |
| 100 | 0.747 | 0.264 | 177.5 | 852.7 |
| 75 | 0.726 | 0.254 | 154.2 | 804.6 |
| 50 | 0.695 | 0.242 | 141.6 | 785.5 |
| 25 | 0.680 | 0.233 | 135.1 | 692.6 |
| 0 (No PLMD) | 0.666 | 0.229 | 0.0 | 573.7 |

Table 14: Ablation for diffusion step $T$ on NVIDIA A40.

| T | IIN SR↑ | IIN SPL↑ | PLMD inference time (s) | Total time (s) |
|---|---------|----------|------------------------|----------------|
| 200 | 0.778 | 0.284 | 84.8 | 419.7 |
| 150 | 0.778 | 0.283 | 74.5 | 372.3 |
| 100 | 0.776 | 0.283 | 68.1 | 327.4 |
| 75 | 0.768 | 0.278 | 64.4 | 315.2 |
| 50 | 0.743 | 0.270 | 61.6 | 304.1 |
| 25 | 0.715 | 0.258 | 60.8 | 298.7 |
| 0 (No PLMD) | 0.702 | 0.252 | 0.0 | 281.6 |

the robot gets stuck. As shown in Fig. 10, most navigation failures are due to *Localization Errors*, which are caused by inaccurate semantic segmentation or incomplete maps. Our PLMD effectively mitigates this issue by generating complete label maps, consistent with the conclusions discussed in Section 4.2.

### A.16 BROADER IMPACTS

Although our training and testing are currently limited to the simulator stage, PLMD can be deployed on real robots. Our PLMD can be seamlessly inserted into mainstream embodied navigation strategies. However, prediction errors in the model may lead to incorrect actions by the robot, potentially causing damage to personal or social property. Therefore, it must be used cautiously to ensure safety in real-world applications.

### A.17 DATA LICENSE

We use three datasets (HM3D_v0.2 (Yadav et al., 2023b), HM3d_v0.1 (Ramakrishnan et al., 2021) and MP3D (Chang et al., 2017)), and employ Habitat simulator. None of these datasets have licenses stated in their official papers or websites. Therefore, we simply cite the corresponding papers without including licenses.

### A.18 THE USE OF LARGE LANGUAGE MODELS (LLMS)

Large Language Models (LLMs) were employed solely as writing aids to polish the language and improve the clarity of expression. They were not used for generating research ideas, designing methods, conducting experiments, or analyzing results. All scientific contributions and substantive content of this work are the sole responsibility of the authors. This use has been disclosed in accordance with the ICLR 2026 policy on LLM usage.

