# OpenReview forum: "Plug-and-Play Label Map Diffusion for Universal Goal-Oriented Navigation"
_ICLR.cc/2026/Conference — Submitted to ICLR 2026_

### Official Review · Reviewer_SDaV · 2025-10-25

**Soundness:** 3
**Presentation:** 3
**Contribution:** 2
**Rating:** 6
**Confidence:** 3

**Summary:**

This paper proposes Plug-and-Play Label Map Diffusion (PLMD), a diffusion-based map completion module to address the "partial observation" problem in Goal-Oriented Navigation (GON). PLMD leverages Denoising Diffusion Probabilistic Models (DDPM) to generate obstacle and semantic labels for unobserved regions, with structural constraints from known obstacles to ensure semantic consistency. It is designed as a plug-and-play module that integrates seamlessly with existing navigation strategies without retraining. Experiments on three GON tasks (ObjectNav, Instance-ImageNav, Multi-Robot ObjectNav) across HM3D and MP3D datasets show PLMD achieves state-of-the-art performance in success rate, path length-weighted success rate, and map completion quality.

**Strengths:**

1. PLMD can be integrated into existing navigation strategies without retraining, reducing the cost of upgrading real-world systems — a key advantage for industrial adoption.

2. Comprehensive validation: The authors test PLMD on three GON tasks (ON/IIN/MRON) across three datasets (HM3D_v0.1/v0.2, MP3D), providing sufficient evidence of cross-task and cross-environment generality.

3. Unlike most simulation-only works, the paper verifies PLMD on Jetson AGX Orin, demonstrating feasibility for resource-constrained embedded robots.

4. Multi-dimensional evaluation: Beyond standard navigation metrics (SR/SPL), the use of PSNR to measure map completion quality links the module’s intermediate performance to final navigation results, enhancing analysis depth.

**Weaknesses:**

1. PLMD’s "obstacle-guided diffusion" is a minor modification of existing diffusion-based map completion (Ji et al., 2024). The paper does not highlight how obstacle constraints solve fundamental limitations of diffusion models in navigation (e.g., mode collapse in sparse environments) — it merely adds a heuristic prior.

2. High computational overhead: PLMD’s FLOPs (34.5G) are an order of magnitude higher than lightweight navigation models (e.g., SemExp’s 3.1G). The paper only mentions "periodic activation" (every 50 steps) as an optimization but does not propose model compression (e.g., quantization, pruning) to improve real-time performance, which is critical for robot navigation. More experiments are required.

3. The authors admit most failures stem from localization errors due to bad semantic segmentation, but PLMD does not include mechanisms to mitigate this (e.g., multi-modal fusion, uncertainty estimation for segmentation). This makes the module fragile in scenarios with low-quality RGB-D inputs.

4. Some related and important works are missing citations: [1] Weakly-Supervised Multi-Granularity Map Learning for Vision-and-Language Navigation [2] IGL-Nav: Incremental 3D Gaussian Localization for Image-goal Navigation [3] Gridmm: Grid memory map for vision-and-language navigation

**Questions:**

See weakness.

---

> ### Author Response · Authors · 2025-11-19
> **Response to Reviewer SDaV**
>
> We appreciate your review and are glad that you recognized the method's practical deployability, comprehensive validation, embedded feasibility, and multi-dimensional evaluation. We are grateful for your positive assessment, and we attempt to respond to your concerns line by line below:
>
> > **W1: The paper does not highlight how obstacle constraints solve fundamental limitations of diffusion models in navigation.**
>
> We appreciate the reviewer raising this point. We formally clarify that the obstacle constraints in PLMD are not heuristically added, but rather designed to address a known limitation of diffusion algorithms in sparse BEV maps. Unlike natural images, BEV maps contain large areas of free space and sparse object pixels. This leads to issues such as room boundary drift and semantic hallucinations in unobserved regions when using purely semantic diffusion methods (e.g., Ji et al [1]). In PLMD, obstacle-aware feature modulation provides stable structural constraints during early semantic diffusion steps when sparse semantic signals are weakest, thereby preventing physically implausible room geometries. Additionally, Ji et al. utilized ground truth maps during training. Our experimental results improved without ground truth maps as training, validating the importance of structural constraints. We elaborate on this point in detail in the Introduction of Rebuttal Revision.
>
> > **W2: High computational overhead**
>
> The number of FLOPs calculated theoretically for PLMD represents only the cost of a single diffusion call and does not occur at every navigation step. In our workflow, PLMD is activated periodically, which empirically aligns with the slow evolution of unobserved map regions. Therefore, PLMD does not impact real-time action. We acknowledge that model compression is an interesting engineering direction, but it is unrelated to the contributions of our paper. Our goal is to demonstrate that injecting structural priors via diffusion significantly improves map completion and subsequent navigation performance, not to design the smallest possible network. Nevertheless, to further address reviewer concerns, we experimented with a compression variant of PLMD. Specifically, we employ ONNX Runtime's static quantization to convert the model into an INT8 ONNX model through QDQ node insertion and MinMax calibration. We observe a reduction in model FLOPs (-24.9) without a significant improvement in computational efficiency. This indicates that navigation latency is primarily dominated by the planner. Although PLMD incurs inference time, it remains acceptable relative to the total navigation time.
>
> |   Component    | IIN SR↑ | IIN SPL↑ | PLMD inference time (s) | Total time (s) | FLOPs (G) | PSNR↑  |
> | :------------: | :-----: | :------: | :---------------------: | :------------: | :-------: | ------ |
> | PLMD-INT8 ONNX |  0.756  |  0.242   |          59.6           |     320.6      |    9.6    | 32.526 |
> |      PLMD      |  0.776  |  0.283   |          68.1           |     327.4      |   34.5    | 34.284 |
>
> > **W3: PLMD does not include mechanisms to mitigate localization errors (e.g., multi-modal fusion, uncertainty estimation for segmentation)**
>
> Thank you for your valuable suggestions. We agree with your perspective. Integrating multi-modal fusion, uncertainty estimation, or pose-aware segmentation into the PLMD framework will be one of our future tasks.
>
> > **W4: Some related and important works are missing citations**
>
> Thank you for your suggestion. We have added references in the related work section.
>
> [1] Ji, Yiming, et al. "Diffusion as reasoning: Enhancing object goal navigation with llm-biased diffusion model." arXiv e-prints (2024): arXiv-2410.

---

### Official Review · Reviewer_trH3 · 2025-10-28

**Soundness:** 3
**Presentation:** 3
**Contribution:** 2
**Rating:** 6
**Confidence:** 5

**Summary:**

This paper proposes a new Plug-and-Play Label Map Diffusion (PLMD) method, which aims to utilize Denoising Diffusion Probabilistic Models (DDPM) to generate a complete obstacle and semantic label map for unobserved regions, enabling goal localization in unknown indoor environments.
Specifically, it designs a label-guide denoising process that leverages obstacle distributions as structural constraints, ensuring consistent and reliable semantic reconstruction at the pixel level.
A clustering algorithm is utilized to identify potential navigation goals in the generated map, forming a candidate goal set.
The map generation is repeated until the candidate goal set is identified or the navigation strategy locates the goal.
Extensive experiments on HM3D and MP3D demonstrate the effectiveness of the proposed method.

**Strengths:**

1)The proposed Label-level Map Completion is delicate and reasonable, which leverages known obstacles and object semantic information in an explicit label map to rebuild unknown regions.
2)The proposed PLMD does not rely on a specific navigation strategy and can be integrated seamlessly into existing navigation strategies that rely on semantic maps.
3)Extensive experiments demonstrate the effectiveness of the proposed method in assisting navigation strategies to locate the goal.

**Weaknesses:**

1) In line 13, Biew -> View.
2) In line 198, the equation to construct the label map dataset does not seem to be fully accurate, since the incomplete label map is stored every 25 steps, t should not be from 0 to F, and the summation symbol is not very appropriate.
3)During training, are the obstacle map network and the semantic map network updated simultaneously after pretraining the obstacle map network? It is not very clear in the current manuscript. Since the obstacle map is first predicted, does it mean the obstacle map is simpler than the semantic map for prediction? Why don't you predict them simultaneously?
4)Why is the time efficiency computed with the method based on LLM/LVM? If LLM/LVM is introduced as the navigation strategy, its potential for predicting goal location should be evaluated and compared, not singly as a navigation strategy. Compared with RL-based methods, the time cost of the proposed diffusion-based method may be too large.
5)The idea of completing a map to predict an object goal is a little similar to PONI, which is not compared in the main manuscript.

**Questions:**

please try to address the weaknesses.

---

> ### Author Response · Authors · 2025-11-19
> **Response to Reviewer trH3**
>
> We are delighted that you consider our method's label-level map completion design to be reasonable, its modular integration a practical strength, and its navigation performance effectively validated through extensive experiments. We are grateful for your positive assessment, and we attempt to respond to your concerns line by line below:
>
> > **W1: In line 13, Biew -> View.**
>
> Thank you for pointing this out. We have addressed it in the Rebuttal Revision.
>
> > **W2: In line 198, the equation to construct the label map dataset does not seem to be fully accurate.**
>
> Thank you for your feedback. We have revised the equation and added more detailed descriptions.
>
> > **W3.1: Are the obstacle map network and the semantic map network updated simultaneously after pretraining the obstacle map network?**
>
> No, they are not updated simultaneously. After pretraining the obstacle map network $\mathcal{G} _ \phi$, we freeze it during semantic map network $\tilde{\mathcal{G}} _ \phi$ training and cease further updates. The obstacle prediction provides a fixed geometric prior, which the semantic generator conditions upon. During initial training, jointly updating both networks caused instability because the geometric prior changed during optimization, leading to room boundary drift and inconsistent obstacle layouts. Freezing the obstacle map network $\mathcal{G} _ \phi$ ensures the semantic generator learns on a stable, physically coherent structural map, resulting in more consistent semantic predictions. We have supplemented Section 3.2 Diffusion Training Process.
>
> > **W3.2:  Since the obstacle map is first predicted, does it mean the obstacle map is simpler than the semantic map for prediction? Why don't you predict them simultaneously?**
>
> Predicting the obstacle map first does not imply a simpler obstacle map, but rather because it encodes the geometric structure that all semantic predictions follow. Directly predicting obstacles and semantics simultaneously leads to unstable diffusion samples, potentially resulting in inconsistent free-space map regions and semantically unreasonable object placements (e.g., chairs inside walls). In our approach, the obstacle map provides a structural prior to constrain geometric shapes; the subsequent semantic diffusion step builds upon this structure to generate spatially coherent and semantically consistent predictions. This hierarchical prediction aligns with the intuition that semantics depend on geometry.
>
> > **W4.1: Why is the time efficiency computed with the method based on LLM/LVM?**
>
>  The time efficiency evaluation involving LLM/LVM planners is not intended to compare their goal location capabilities against PLMD. LLM/LVM timing data is included because they serve as planning/decision modules within the navigation pipeline, and as advanced planners, their latency characteristics differ significantly from reinforcement learning-based agents. **Our core objective is to analyze whether the diffusion-based PLMD prediction mechanism can accommodate the overall computational budget of modern planners** (See **W4.2**). In the Rebuttal Revision, we additionally quantified the computational time efficiency of the ON/MRON (SemExp) reinforcement learning-based approach and supplemented Section A.8 (values in parentheses indicate the proportion of time spent on PLMD).
>
> | **Task** | PLMD inference time (s) |   Total time (s)   |
> | :------: | :---------------------: | :----------------: |
> |    ON    |     139.1 (15.95%)      | 872.1 (with LLMs)  |
> |    ON    |      52.9 (20.15%)      |       262.5        |
> |   MRON   |      120.5 (9.66%)      | 1247.0 (with LLMs) |
> |   MRON   |      79.7 (19.67%)      |       405.2        |
> |   IIN    |      68.1 (20.80%)      |       327.4        |
>
> > **W4.2: Compared with RL-based methods, the time cost of the proposed diffusion-based method may be too large.**
>
> PLMD does not replace navigation strategies; it provides enhanced map completion capabilities for downstream planners while also enabling direct prediction of positioning goals. Navigation latency is dominated by the planner, and we observe that PLMD accounts for 10–20% of total inference time (even in RL), remaining acceptable compared to the total navigation time. The diffusion step in PLMD introduces only a small fixed overhead and does not accumulate computational costs during long-term planning.
>
> > **W5: The idea of completing a map to predict an object goal is a little similar to PONI, which is not compared in the main manuscript.**
>
> Thank you for your suggestion. We have included the PONI comparison in Table 1 of our main manuscript.

---

### Official Review · Reviewer_bH3k · 2025-10-29

**Soundness:** 1
**Presentation:** 2
**Contribution:** 1
**Rating:** 0
**Confidence:** 5

**Summary:**

This paper addresses the problem of navigating to a goal in an unknown environment. The paper uses diffusion to generate a prediction of the structure of the unknown part of the map for the purposes of navigation. The paper shows evaluations on Habitat-Matterport3D and Matterport3D and show that the approach compares favourably to a set of baselines.

**Strengths:**

The paper addresses an interesting and important problem in robotics, which is goal-directed navigation in unknown or partially unknown environments. The idea of using diffusion to predict the unknown parts of the map is interesting, and worth studying.

**Weaknesses:**

Unfortunately, there is very little to recommend this paper for acceptance. The primary objection I have is that the training process for the diffusion model appears to be entirely in the same environment as the test evaluation, as given by "we generate training and validation data from label maps collected during the interaction of robot with the environment. First, we randomly initialize a starting position within indoor environments" and "To train the PLMD, we collect obstacle and semantic maps of size 256×256 through the Habitat simulator". Allowing the system to go and build a diffusion model of the environment is fundamentally no different (and arguably harder) than allowing the robot to go and build a map ahead of time. This paper could only be of interest if there had been any attempt to test in one environment and evaluate in another, different environment. The authors do not even appear to have partitioned the environment into disjoint train and test areas.

Secondly, the experimental results are very unclear. How is the map updated?  What (simulated) sensor data is used? It is not clear why success is not 100% -- if the goal object is not at the predicted location, does the mission terminate? That is not a particularly useful setting for this problem -- a far more useful (and common) setting is where diffusion model is used as a prior, and expected costs are calculated using the model as a prior, iterating via replanning until the goal is found.

The paper focuses on finding goals in unknown maps, but does not compare against the fairly large literature in robotics that addresses that problem, and primarily compares itself against RL and other diffusion approaches. It would be interesting to compare against the planning under uncertainty work that leverages structured models (e.g., work by Greg Stein, Nikolay Atanasov, Luca Carlone, etc.).

If the paper had focused entirely on the problem of map prediction and not the navigation problem, I might be more in favour of this paper, but even viewed in that light, the contribution is relatively modest. The paper presents results that indicate that the two-level label map captures "the contextual relationship between obstacles and semantic features" but this seems to be a statement that explicitly modelling the relationship between obstacles and semantics provides a better model overall, which has been known for sometime in the semantic SLAM community.

**Questions:**

- Why does the robot not succeed 100% of the time? Do the trials terminate early?
- How well does the approach generalise across environments? Can it be trained in one environment and succeed in another?

---

> ### Author Response · Authors · 2025-11-19
> **Response to Reviewer bH3k - Part 1**
>
> Thank you for your detailed review. We appreciate your recognition of the importance and interest of the problem addressed, as well as the value and worth of studying employing diffusion models for map prediction. However, we believe the reviewer may have developed some misunderstandings of our work. First, **we recommend that the reviewer RE-READ and RE-EVALUATE this paper.** If the reviewer still has concerns after careful review, please allow us to address them line by line below:
>
> >  **W1: Training and testing in the same environment.**
>
> **SubW1.1: Training process for the diffusion model appears to be entirely in the same environment as the test evaluation.**
>
> We recommend that reviewers carefully read Sections A.3, A.5 and A.6 of our paper. In Section A.5, we divided the diverse label map observation pairs in the HM3D_v0.1 dataset into separate training and validation sets with corresponding scenes. We trained the diffusion model on the training set and conducted navigation validation on the validation set, ensuring no overlap between training and validation samples. To prevent further misunderstanding, we have added a more detailed description in Section 3.2 Label Map Data Collection, Section A.3 and Section A.5 of the Rebuttal Revision. Additionally, in Section A.6, we conducted cross-dataset evaluations to demonstrate that PLMD maintains robustness across different environmental distributions.
>
> **SubW1.2: Allowing the system to go and build a diffusion model of the environment is fundamentally no different (and arguably harder) than allowing the robot to go and build a map ahead of time.**
>
> The reviewer's statement that "Allowing the system to go and build a diffusion model of the environment is fundamentally no different (and arguably harder) than allowing the robot to go and build a map ahead of time" seems to misinterpret our original intent. Please refer to Abstract line 25 and Introduction lines 92-93 of our paper. PLMD aims to expand the region of the unknown map, providing a broader Bird’s-Eye View map perception domain for navigation strategies. Our emphasis is on predicting and completing maps, requiring robots to conduct preliminary observations, not the reviewer’s interpretation of "allowing the robot to go and build a map ahead of time."
>
> **SubW1.3: This paper could only be of interest if there had been any attempt to test in one environment and evaluate in another, different environment.**
>
> We recommend that the reviewer carefully read Section A.6 of our paper. We further trained PLD on the MP3D dataset under identical configurations as in Section 4.1 and Section A.5 (we omitted HM3D_v0.2 due to its scene similarity with HM3D_v0.1). Cross-validation on HM3D_v0.2 and MP3D datasets demonstrated PLMD's robust out-of-distribution performance across diverse environments.
>
> > **W2: Unclear experimental protocol and unrealistic navigation setting**
>
> **SubW2.1: How is the map updated? What (simulated) sensor data is used?**
>
> We recommend that reviewers carefully read Sections 3.1 and 3.2 of our paper. Section 3.1 details our method for updating the label map. Specifically, at each time step, the robot constructs and updates a self-centered global semantic map by fusing RGB-D observation sensor data with pose sensor data. Furthermore, we present a detailed prediction map update method in Section 3.2 and Figure 2 (a) of the paper. Our pre-trained diffusion models then intelligently inpaint these masked regions. The resulting predicted map vectors are merged back into the global map $M_t$, replacing unknown areas with plausible predictions.
>
> **SubW2.2: If the goal object is not at the predicted location, does the mission terminate?**
>
> We recommend that reviewers carefully read Section 3.3 and Section A.3 of our paper. In Section 3.3, we decouple PLMD-assisted navigation into Label Map Restored and Localization Strategy. In Section A.3, we defined termination conditions for each task (e.g., ON has a maximum time step limit of 500 per epoch; if steps exceed 500, the task fails). The former replaces the original map, while the latter is implemented in line 298: "If no valid clustering can be found within the region, the navigation strategy is executed." We believe this will address the reviewer's concern.
>
> **SubSuggestion1: A far more useful (and common) setting is where diffusion model is used as a prior, and expected costs are calculated using the model as a prior, iterating via replanning until the goal is found.**
>
> Thank you for your suggestion. In fact, the reviewer's suggestion is one of the core aspects of our work. As the reviewer noted, PLMD serves as the "prior" for the navigation strategy, continuously generating prediction maps based on the label map of the current time step during the navigation strategy's execution until the goal is located.

---

> > ### Author Response · Authors · 2025-11-19
> > **Response to Reviewer bH3k - Part 2**
> >
> > > **W3: Lack of comparison with relevant robotics literature.**
> >
> > We appreciate the reviewer's suggestion. The methods cited by the reviewers are valuable for global trajectory planning and uncertainty management across entire navigation paths [1-3]. However, our approach focuses on local, incremental map prediction designed to enhance navigation performance without requiring explicit uncertainty models. Thus, we did not include these approaches in our comparison, as they do not directly address the core of our method: local map completion and goal localization.
> >
> > > **W4: Explicitly modelling the relationship between obstacles and semantics provides a better model overall.**
> >
> > Unlike explicit modeling SLAM, our approach focuses on goal-oriented navigation, enabling robots to not only utilize map completion for localization but also assist in goal localization within unobserved regions. While semantic SLAM methods typically emphasize creating complete maps for localization, our approach employs diffusion models to predict unobserved regions in real time, enabling robots to navigate toward specific targets even when the environment is partially visible.
> >
> > > **Q1: Why does the robot not succeed 100% of the time? Do the trials terminate early?**
> >
> > Please see **SubW2.2**.
> >
> > > **Q2: How well does the approach generalise across environments? Can it be trained in one environment and succeed in another?**
> >
> > Please see **SubW1.3**.
> >
> >
> >
> > [1] Paudel, Abhishek, Xuesu Xiao, and Gregory J. Stein. "Multi-Strategy Deployment-Time Learning and Adaptation for Navigation under Uncertainty." 8th Annual Conference on Robot Learning. 2024.
> >
> > [2] Ostertag, Michael, Nikolay Atanasov, and Tajana Rosing. "Trajectory planning and optimization for minimizing uncertainty in persistent monitoring applications." Journal of Intelligent & Robotic Systems 106.1 (2022): 2.
> >
> > [3] Carlone, Luca, and Daniel Lyons. "Uncertainty-constrained robot exploration: A mixed-integer linear programming approach." 2014 IEEE International Conference on Robotics and Automation (ICRA). IEEE, 2014.

---

### Official Review · Reviewer_3tou · 2025-10-31

**Soundness:** 3
**Presentation:** 3
**Contribution:** 2
**Rating:** 6
**Confidence:** 4

**Summary:**

This paper proposes PLMD (Plug-and-Play Label Map Diffusion), a diffusion-based framework for semantic and obstacle map completion to enhance embodied navigation.
- The method performs diffusion inference under joint conditioning of semantic and obstacle labels for structural map completion and introduces a clustering-based integration strategy for selecting long-range navigation goals.
- Experiments are conducted on the HM3D dataset, covering ObjectNav (ON), ImageNav (IIN), and Multi-Robot ObjectNav (MRON) tasks.

**Strengths:**

- The model can be seamlessly integrated into existing navigation systems, demonstrating strong modularity and generality.
- PLMD achieves performance improvements across multiple subtasks (ON, IIN, MRON), and the generated maps provide intuitive visual evidence.

**Weaknesses:**

- The integration strategy for cluster core selection relies on fixed, manually tuned weights (0.5 / 0.4 / 0.1) that linearly combine cluster size, semantic confidence, and distance penalty. However, the paper does not provide any theoretical justification or empirical validation for this formulation. Such empirically chosen parameters may overfit to specific datasets (e.g., HM3D), limiting generalization and reproducibility. It is recommended to include a weight sensitivity study, showing how different combinations affect SR/SPL performance, and to clarify the individual contributions of each component to navigation quality.
- Diffusion-based models typically involve lengthy iterative inference, and PLMD requires optimal reconstruction under the joint conditioning of obstacle and semantic labels, which may further increase inference time. However, the paper does not report the model’s average inference time, computational resource consumption, or its impact on real-time performance during online navigation. The absence of these measurements limits the assessment of the method’s engineering practicality and deployability. It is recommended to include additional validation experiments reporting average inference latency and GPU resource usage to demonstrate the real-time feasibility of the proposed approach in embodied intelligence scenarios.
- The paper claims that PLMD is a “general-purpose navigation enhancement method,” but all experiments are conducted primarily on closed-set semantic datasets. The model has not been evaluated on open-vocabulary tasks, zero-shot or few-shot transfer scenarios, nor has its performance been verified across different datasets or real-world environments. This weakens the evidence supporting the claim of generality. Since real-world environments typically contain open semantics and unknown structures, the current experiments do not demonstrate the model’s actual adaptability. It is recommended to conduct evaluations on open-vocabulary tasks or cross-dataset transfer experiments to substantiate the core claim of “general navigation.”

**Questions:**

- The idea of using generative models for map completion in goal-oriented navigation is not new — for example, “Imagine Before Go” and “Distilling LLM Prior to Flow Model” have explored similar directions.
Please clarify how PLMD differs from these works in terms of the generation mechanism, type of prior, and the key insight or advantage it brings beyond existing generative map completion methods.
- Why were OpenFMNav, FBE, and MCoCoNav chosen as the base navigation models instead of other alternatives?
- How was the choice of 100 diffusion steps determined? Was it selected as a trade-off between inference time and generation quality?

---

> ### Author Response · Authors · 2025-11-19
> **Response to Reviewer 3tou - Part 1**
>
> We are delighted that you consider our method to be modular, generally effective across multiple navigation tasks, and provide intuitive visual evidence for its predictions. We are grateful for your positive assessment, and we attempt to respond to your concerns line by line below:
>
> > **W1: Critique of Fixed Weights in Cluster Core Selection**
>
> Thank you for your valuable comments and suggestions. We evaluated the performance of five representative weight distributions: 1) our initial design (0.5/0.4/0.1), 2) density-dominant (0.7/0.2/0.1), 3) size-dominant (0.2/0.7/0.1), 4) distance-dominant (0.1/0.2/0.7), and 5) uniform distribution (0.33/0.33/0.34). Due to time constraints during the rebuttal phase, we randomly selected 200 episodes each from HM3D_v0.2 and MP3D as MRON validation subset. Across all test configurations, the original ratio (0.5/0.4/0.1) consistently achieved the best SR and SPL performance on both datasets. This indicates that the weighting scheme does not overfit HM3D and demonstrates strong generalization capabilities across diverse environments. Cluster density and size contribute more directly to navigation success than distance to the starting point. We have supplemented this finding in the Rebuttal Revision.
>
> | Weight Distributions | HM3D SR↑  | HM3D SPL↑ | MP3D SR↑  | MP3D SPL↑ |
> | :------------------: | :-------: | :-------: | :-------: | :-------: |
> |     0.5/0.4/0.1      | **0.726** | **0.401** | **0.588** | **0.382** |
> |     0.7/0.2/0.1      |   0.721   |   0.395   |   0.585   |   0.382   |
> |     0.2/0.7/0.1      |   0.711   |   0.398   |   0.552   |   0.344   |
> |     0.1/0.2/0.7      |   0.707   |   0.385   |   0.537   |   0.351   |
> |    0.33/0.33/0.34    |   0.715   |   0.384   |   0.572   |   0.369   |
>
> > **W2: Inference Time and Computational Efficiency Analysis for PLMD**
>
> In Section A.7, A.8, and A.11, we provide detailed information on PLMD efficiency, including FLOPs, number of parameters, GPU resource usage on A40 and Jetson AGX Orin, and the design rationale behind periodic activation (every 50 steps). These metrics confirm that PLMD introduces only minimal overhead relative to the planner itself and can be easily integrated into real-time embodied navigation systems.
>
> > **W3: Generalizability Evaluation of PLMD as a General-Purpose Navigation Method**
>
> Thank you for your valuable suggestions. We employ the term "general navigation enhancement" to denote PLMD's task-level generality, meaning it can improve performance across different navigation paradigms without altering existing architectures. Furthermore, PLMD has undergone cross-validation across multiple datasets, including MP3D and HM3D_v0.2, demonstrating that structural and semantic prior information can be transferred across different datasets. We expanded the evaluation scope and retrained the PLMD model using an open-vocabulary segmentation process based on Grounded SAM [1]. Our revised paper adds an open-vocabulary evaluation, where we additionally selected three object categories (lamp, toy car, microwave) not present in any closed vocabulary set and conducted the most representative MRON experiments on the HM3D_v0.2 validation subset extracted from **W1**. PLDM* denotes a variant using Grounded SAM as the semantic segmenter. While the original closed vocabulary segmentation module performed poorly on these out-of-distribution categories, the Grounded SAM-based PLMD demonstrated improved performance, confirming that our obstacle-based diffusion architecture can indeed transfer to open-word goals. We also note that Grounded SAM significantly increases the overall navigation time. Furthermore, as described in Section A.6, PLMD was trained on the MP3D dataset and evaluated on unseen HM3D_v0.2 navigation tasks, demonstrating its cross-dataset generalization capabilities for both structural and semantic priors. We will conduct broader open-vocabulary evaluations and real-world robotic experiments following the rebuttal phase.
>
> | Method       | SR↑   | SPL↑  | Total time (s) |
> | :----------- | :---- | ----- | -------------- |
> | Multi-SemExp | 0.317 | 0.222 | 246.8          |
> | MCoCoNav     | 0.399 | 0.265 | 1007.1         |
> | PLMD         | 0.363 | 0.232 | 1169.2         |
> | PLMD*        | 0.446 | 0.274 | 1540.2         |

---

> ### Author Response · Authors · 2025-11-19
> **Response to Reviewer 3tou - Part 2**
>
> > **Q1: Clarification on PLMD's Differentiation from Existing Generative Map Completion Methods**
>
> First, Zhang et al. [2] and Li et al. [3] rely on high-level textual priors provided by LLMs. Their predictions often lack geometric constraints and fail to maintain the continuity of navigable free space (as LLMs inherently lack the ability to understand the topological structure of physical space). In contrast, PLMD employs an obstacle map network to first reconstruct structural geometry (walls, room boundaries, free space topology), while a semantic map network modulates this through spatial adaptation, conditioned on multi-scale obstacle features. Furthermore, PLMD contributes a key insight absent in prior generative navigation methods: explicit geometric conditioning is crucial for stable, navigation-relevant generative map completion. Finally, PLMD's core advantage lies in generating navigation-effective, obstacle-consistent map completions while preserving room layout and passability. We add clearer comparisons in the related work section to highlight these distinctions.
>
> > **Q2: Why were OpenFMNav, FBE, and MCoCoNav chosen as the base navigation models instead of other alternatives?**
>
> OpenFMNav, FBE, and MCoCoNav represent three distinct yet widely applicable goal-oriented navigation planners. 1) OpenFMNav is a novel planner based on LLM/VLM, embodying an emerging foundation model-based navigation framework. 2) FBE serves as a robust classical benchmark revealing pure exploration advantages (without requiring learned policies). 3) MCoCoNav addresses multi-robot collaborative planning. Together, these three baseline approaches encompass foundational models, classical geometric, and multi-robot navigation paradigms. Furthermore, our method incorporates additional learning-based planning baselines, including IEVE [4], ensuring PLMD can be evaluated across diverse planning mechanisms rather than being tailored for any single planner type.
>
> > **Q3: How was the choice of 100 diffusion steps determined? Was it selected as a trade-off between inference time and generation quality?**
>
> Yes, this choice stems from a practical trade-off between reconstruction quality and inference latency, while also being informed by experience with DDPM. We conducted additional ablation studies on Jetson Orin with reduced diffusion steps (T = 25, 50, 75) and higher diffusion steps (T = 150, 200). The results demonstrate a trade-off between navigation performance and computational cost: while reducing T from 200 to 100 decreases IIN SR by only 0.5%, it achieves a 31.4% reduction in total time (from 1242.3s to 852.7s); however, further reducing T (from 100 to 25) leads to a more significant 6.7% drop in IIN SR despite greater time savings. The optimal balance occurs at T = 100, where navigation performance peaks with acceptable time consumption.
>
> | T           | IIN SR↑ | IIN SPL↑ | PLMD inference time (s) | Total time (s) |
> | :---------- | :------ | ------------ | ----------------------- | -------------- |
> | 200         | 0.752   | 0.270        | 261.8                   | 1242.3         |
> | 150         | 0.750   | 0.268        | 222.2                   | 1085.7         |
> | 100         | 0.747   | 0.264        | 177.5                   | 852.7          |
> | 75          | 0.726   | 0.254        | 154.2                   | 804.6          |
> | 50          | 0.695   | 0.242        | 141.6                   | 785.5          |
> | 25          | 0.680   | 0.233        | 135.1                   | 692.6          |
> | 0 (No PLMD) | 0.666   | 0.229        | 0.0                     | 573.7          |
>
>
>
> [1] Ren, Tianhe, et al. "Grounded sam: Assembling open-world models for diverse visual tasks." arXiv preprint arXiv:2401.14159 (2024).
>
> [2] Zhang, Sixian, et al. "Imagine before go: Self-supervised generative map for object goal navigation." Proceedings of the IEEE/CVF Conference on Computer Vision and Pattern Recognition. 2024.
>
> [3] Li, Badi, et al. "Distilling LLM Prior to Flow Model for Generalizable Agent's Imagination in Object Goal Navigation." arXiv preprint arXiv:2508.09423 (2025).
>
> [4] Lei, Xiaohan, et al. "Instance-aware exploration-verification-exploitation for instance imagegoal navigation." Proceedings of the IEEE/CVF Conference on Computer Vision and Pattern Recognition. 2024.

---

> ### Author Response · Authors · 2025-11-29
> **Response to Reviewer 3tou - Part 3 (Full experiments)**
>
> We once again sincerely thank the reviewer for your valuable comments. In this response, we provide supplementary experiments for **W1** and **W3** using the **Full HM3D_v0.2 and MP3D datasets**. Additionally, we supplement ablation experiments for Q3 on NVIDIA A40.
>
> We found that the Full datasets did not yield significantly different results compared to the subset. This finding has been incorporated into Section 4.2, while the original subset experiments are now detailed in Appendix A.12 and A.13.
>
> > **W1: Critique of Fixed Weights in Cluster Core Selection**
>
> | Weight Distributions | HM3D SR↑  | HM3D SPL↑ | MP3D SR↑  | MP3D SPL↑ |
> | :------------------: | :-------: | :-------: | :-------: | :-------: |
> |     0.5/0.4/0.1      | **0.762** | **0.406** | **0.591** | **0.382** |
> |     0.7/0.2/0.1      |   0.758   |   0.399   |   0.588   |   0.382   |
> |     0.2/0.7/0.1      |   0.743   |   0.392   |   0.562   |   0.363   |
> |     0.1/0.2/0.7      |   0.740   |   0.385   |   0.540   |   0.366   |
> |    0.33/0.33/0.34    |   0.741   |   0.385   |   0.577   |   0.375   |
>
> > **W3: Generalizability Evaluation of PLMD as a General-Purpose Navigation Method**
>
> | Method       | SR↑   | SPL↑  | Total time (s) |
> | :----------- | :---- | ----- | -------------- |
> | Multi-SemExp | 0.285 | 0.206 | 263.5          |
> | MCoCoNav     | 0.327 | 0.242 | 1063.6         |
> | PLMD         | 0.323 | 0.225 | 1195.7         |
> | PLMD*        | 0.354 | 0.268 | 1535.6         |
>
> > **Q3: How was the choice of 100 diffusion steps determined? Was it selected as a trade-off between inference time and generation quality?**
>
> Compared to results on Jetson Orin, performance on A40 shows further improvements. We have supplemented these results along with the runtime for both A40 and Jetson Orin platforms in Section A.14.
>
> | T           | IIN SR↑ | IIN SPL↑ | PLMD inference time (s) | Total time (s) |
> | :---------- | :------ | ------------ | ----------------------- | -------------- |
> | 200         | 0.778   | 0.284        | 84.8                    | 419.7          |
> | 150         | 0.778   | 0.283        | 74.5                    | 372.3          |
> | 100         | 0.776   | 0.283        | 68.1                    | 327.4          |
> | 75          | 0.768   | 0.278        | 64.4                    | 315.2          |
> | 50          | 0.743   | 0.270        | 61.6                    | 304.1          |
> | 25          | 0.715   | 0.258        | 60.8                    | 298.7          |
> | 0 (No PLMD) | 0.702   | 0.252        | 0.0                     | 281.6          |

---

### Author Response · Authors · 2025-11-19
**Overall response**

Dear PCs, SACs, ACs and Reviewers,

Thank you very much for your valuable comments and insightful suggestions. During the rebuttal period, we follow your suggestions and try to deal with your concerns from the following aspects:

1. In response to reviewer **3tou**'s feedback, we conducted three additional experiments: First, we evaluated five weight distributions on HM3D and MP3D, confirming that the original weights of 0.5/0.4/0.1 remain optimal for both SR and SPL. Second, we supplemented experiments on PLMD's generalization to open-vocabulary goals, finding that replacing the original closed-vocabulary semantic segmentation tool with Grounded SAM improves navigation performance for unseen categories (e.g., lamp, toy car). Finally, ablation experiments with varying diffusion steps confirmed T=100 achieves the optimal balance between inference time and generation quality, reducing total time while maintaining navigation performance.
2. Based on reviewer **bH3K**'s feedback, we refined descriptions of the training and validation sets in Sections 3.2, A.3, A.5, and A.6.
3. Following reviewer **trH3**'s suggestions, we improve the writing of the paper. Section 3.2 now includes additional details on training the obstacle map network. Additionally, we added computational efficiency analysis for reinforcement learning methods like ON/MRON. Section A.8 now include time distribution data for PLMD across RL-based ON/MRON tasks, demonstrating its computational overhead remains within acceptable limits.
4. In response to reviewer **SDaV**'s feedback, we clarified the importance of obstacle constraints for pure semantic diffusion methods in sparse BEV maps within the introduction. Then we conduct model quantization experiments. Results show that quantized FLOPs are significantly reduced without notable computational efficiency gains, while navigation performance and map quality deteriorate, confirming that navigation latency is primarily determined by the planner.
5. Following suggestions from reviewers **3tou, trH3, and SDaV,** we further refined theoretical clarifications and elaborations in related work.

Finally, we would like to thank the four anonymous reviewers once again for their invaluable comments and suggestions, which substantially enhanced the technical quality and clarity of our manuscript. We also thank the PCs, SACs and ACs for revisiting our response.

Sincerely,

Authors of Paper 7200

---

### Meta-Review · Area_Chair_Squd · 2026-01-04

**Summary:**

The submission initially received polarized reviews: three reviewers rated it borderline accept / weak accept, while one reviewer issued a strong reject.

**What the paper does (consensus understanding):**
- Proposes PLMD, a plug-and-play diffusion-based map completion module for Goal-Oriented Navigation (GON).
- Uses DDPMs to jointly complete obstacle and semantic label maps in unobserved BEV regions.
- Treats diffusion as a structural prior for expanding unknown map regions during navigation.
- Integrates with existing planner without retraining.
- Evaluated on HM3D and MP3D, including embedded hardware (Jetson AGX Orin).

**Broadly agreed strengths:**
- Modularity / plug-and-play design.
- Strong experimental coverage across tasks, planners, datasets, and hardware.
- Practical motivation: low-cost upgrade to existing navigation systems.
- Obstacle-guided diffusion improves structural consistency in sparse maps.

**Core axes of disagreement:**
- Whether PLMD truly generalizes or is effectively trained in-distribution.
- Whether diffusion-based map completion is conceptually novel.
- Whether the engineering cost (latency, FLOPs) is justified.
- Whether the paper sufficiently engages with classical planning-under-uncertainty literature.

**Reviewer Concerns:**

## Reviewer concerns addressed


**1. Fixed weights in cluster-core selection:**
Heuristic clustering weights may overfit a single dataset.

**Rebuttal:**
- Added weight sensitivity studies across HM3D and MP3D.
- Evaluated multiple weighting strategies.


**2. Inference time and deployability:**
Diffusion is expensive and may not be real-time.

**Rebuttal:**
- Added detailed runtime, FLOPs, and hardware benchmarks.
- Showed PLMD contributes only 10–20% of total navigation time.
- Demonstrated feasibility on Jetson AGX Orin.


**3. Generalization across datasets and open vocabulary:**
“Universal navigation” claim unsupported.

**Rebuttal:**
- Added cross-dataset evaluation.
- Added open-vocabulary goals using Grounded SAM.
- Showed consistent SR/SPL improvements on unseen categories.

**4. Choice of diffusion steps:**
Arbitrary diffusion step selection.

**Rebuttal:**
- Added ablation over T = 25-200.
- Identified a clear performance-latency trade-off.



**5. Training protocol clarity:**
Potential data leakage or unclear splits.

**Rebuttal:**
- Clarified scene-disjoint splits.
- Explicitly described cross-dataset validation.



**6. Missing comparisons and citations:**
Missing key baselines and related work.

**Rebuttal outcome:**
- Added PONI comparison.
- Expanded related work.

---
## Reviewer concerns that remain partially or fully outstanding

**1. Conceptual framing disagreement (strong reject)**
- Reviewer argues the contribution of map prediction is modest.


**2. Novelty relative to prior diffusion-based map completion**
- Obstacle-guided diffusion may be seen as incremental.


**3. Fragility to semantic segmentation errors**
- Failures often trace back to poor perception.

**4. Missing comparison to classical planning-under-uncertainty**
- No evaluation against belief-space or uncertainty-aware planners.

**Reviewer Scores:**

This paper received initial scores 0, 6, 6, 6. The 6s may be kept or increased. The 0 score will high-likely be increased as some key concerns are addressed, but may not reach a positive level due to some outstanding key concerns, such as comparison to the broad navigation methods. Another key weakness is incomplete comparison, even with currently chosen methods, such as missing results of baseline methods (eg, FBE, OpenFMNav). Last but not least, the performance improvement of the main metric SPL is marginal on ON (unclear on the other two benchmarks due to the lack of results of many methods). This indicates that the improved SR mainly comes from increased exploration (more navigation steps), which is inefficient.

---

### Decision · Program_Chairs · 2026-01-26

Reject